# Environmental enrichment preserves a young DNA methylation landscape in the aged mouse hippocampus

Sara Zocher [1,2], Rupert W. Overall[1,2], Mathias Lesche [2,3], Andreas Dahl [2,3] & Gerd Kempermann [1,2✉]

The decline of brain function during aging is associated with epigenetic changes, including DNA methylation. Lifestyle interventions can improve brain function during aging, but their influence on age-related epigenetic changes is unknown. Using genome-wide DNA methylation sequencing, we here show that experiencing a stimulus-rich environment counteracts age-related DNA methylation changes in the hippocampal dentate gyrus of mice. Specifically, environmental enrichment prevented the aging-induced CpG hypomethylation at target sites of the methyl-CpG-binding protein Mecp2, which is critical to neuronal function. The genes at which environmental enrichment counteracted aging effects have described roles in neuronal plasticity, neuronal cell communication and adult hippocampal neurogenesis and are dysregulated with age-related cognitive decline in the human brain. Our results highlight the stimulating effects of environmental enrichment on hippocampal plasticity at the level of DNA methylation and give molecular insights into the specific aspects of brain aging that can be counteracted by lifestyle interventions.

[1] German Center for Neurodegenerative Diseases (DZNE) Dresden, Dresden, Germany. [2] Center for Regenerative Therapies Dresden (CRTD), Technische Universität Dresden, Dresden, Germany. [3] DRESDEN-concept Genome Center c/o Center for Molecular and Cellular Bioengineering (CMCB), Technische Universität Dresden, Dresden, Germany. ✉email: Gerd.Kempermann@dzne.de

Aging is associated with a progressive decline in brain function that manifests in cognitive impairments, increased risk for neurodegenerative diseases, and loss of neural plasticity. Lifestyle factors, including physical exercise, cognitive stimulation, and social interaction, attenuate age-related reductions of brain function in humans[1–4], contributing to what is called "reserve" or "maintenance" of brain function[5,6]. In rodent models, environmental enrichment (ENR) has repeatedly demonstrated the beneficial effects of a stimulus-rich environment and behavioral activity on life-long brain plasticity and brain health[7–9]. In enriched environments, groups of rodents freely explore large cages equipped with frequently rearranged toys, providing physical, cognitive, sensory, and social stimulation[10]. ENR promotes the function of existing neurons and adult neurogenesis in the hippocampus[11,12]—a brain region with key roles in learning and memory, but high susceptibility to stress- and age-related impairments[13]. ENR-stimulated brain plasticity has been linked to numerous behavioral changes, including reduced anxiety, improved motor coordination, altered exploratory activity, and enhanced cognition[14–16]. In mouse models of age-related neurodegenerative diseases, ENR attenuated disease progression and ameliorated behavioral deficits after disease onset[17,18]. Similarly, aged rodents and animal models of accelerated aging showed improved learning and enhanced brain plasticity when housed in ENR[19–24]. How ENR and lifestyle factors counteract aging effects at the molecular level is largely unknown, but could provide potential therapeutic targets to combat age-related cognitive decline[9]. Therefore, we set out to investigate the molecular mechanisms that link environmentally induced brain plasticity with improved brain maintenance during aging.

Genome-wide DNA methylation changes are hallmarks of cellular aging[25] and are used as biomarkers of the aging process[26]. In the hippocampus, the aging-induced dysregulation of the DNA methylation machinery is involved in the development of cognitive impairments[27,28]. Proper control of neuronal DNA methylation patterns in the adult brain is crucial for dynamic gene expression changes associated with synaptic plasticity and memory formation[29–31]. The sensitivity of neuronal methylomes to environmental stimuli, such as maternal behavior[32], learning stimuli[33], or isolated neuronal activation[34], contributes to the molecular mechanisms underlying experience-dependent brain plasticity[35]. Conversely, aberrant changes of neuronal DNA methylation patterns have been described in aging[36,37] and in age-related disorders, such as Alzheimer's disease[38,39]. Reversal of age-related DNA methylation changes has been proposed to be a fundamental mechanism of interventions that delay aging-induced tissue degeneration[40]. Since neuronal DNA methylation patterns are plastic and sensitive to environmental experiences, behavioral interventions could potentially rescue aberrant DNA methylation changes and thereby promote brain health in old age. Previous studies already suggested that ENR can modify epigenetic modifications, including DNA methylation, in the brain[41–43], but whether those changes interact with age-related DNA methylation changes remained unexplored.

In this work, we investigate the influence of ENR on DNA methylation patterns in the hippocampal dentate gyrus and find that ENR counteracts aging-induced DNA methylation changes at genes related to neuronal plasticity and adult hippocampal neurogenesis. Our results highlight the potential of lifetime experiences to influence brain health in old age and provide a possible mechanism underlying the effects of lifestyle factors on brain aging.

## Results

**ENR modifies DNA methylation patterns in the adult dentate gyrus.** To first investigate whether ENR changes DNA methylation patterns in the adult dentate gyrus, we performed a pilot experiment

and kept female C57BL/6JRj mice in ENR or standard housing cages (STD) for three months starting at an age of six weeks. For the particular ENR paradigm used here, our previous studies had already confirmed the stimulating effect on adult hippocampal neurogenesis and hippocampus-dependent memory and learning[15,16]. We performed genome-wide DNA methylation profiling on micro-dissected dentate gyrus tissue of three animals per group by reduced representation bisulfite sequencing (RRBS)[44,45]. No ENR-induced changes in global CpG methylation levels were detected (Fig. 1a), underscoring general genomic stability after ENR. Nevertheless, ENR modified methylation levels at 11,101 individual CpGs, i.e., 1.25% of all covered CpGs (Fig. 1b). Differentially methylated CpGs (dmCpGs) were depleted at CpG islands, CpG island shores, promoters, and exons but enriched at enhancers, introns, and intergenic regions of the genome (Fig. 1c). In addition, ENR changed methylation at 0.019% of CpHs (in total only 750 differentially methylated CpHs; dmCpHs), which showed a similar genomic distribution to dmCpGs but no enrichment at enhancers (Supplementary Fig. 1; Supplementary Data 1). Thus, ENR changed methylation in the dentate gyrus predominantly at CpGs located within regulatory genomic regions.

To explore the neuronal processes that are regulated by ENR, we performed gene set enrichment analysis with the 373 ENR-induced differentially methylated genes using expert-curated knowledge bases. Gene ontology (GO)[46] and Reactome pathway analyses[47] showed that ENR-induced dmCpGs were enriched at genes involved in structural components of neurons, such as "axon part", "dendrite", and "dendritic spine" (Fig. 1d), and at genes with known functions in synaptic plasticity pathways, including glutamate receptor signaling and axon guidance (Fig. 1e; Supplementary Fig. 2; Supplementary Data 2). Enrichment analysis for genes from the mammalian adult neurogenesis gene ontology (MANGO)[48] highlighted that ENR changed DNA methylation at genes with described function in hippocampal neurogenesis, such as *Fgfr1*, *Gria1*, *Nfatc4*, *Ntf3*, *Flt1*, and *Thrb* (Fig. 1f). In addition, enrichment analysis using the synaptic gene ontologies (SynGO) knowledgebase[49] suggested that ENR-induced differentially methylated genes are enriched at genes involved in synaptic assembly, organization of post-synapses, and neurotransmitter signaling (Fig. 1g). These results indicated that ENR regulates pathways involved in neuronal plasticity and adult hippocampal neurogenesis in the murine dentate gyrus.

To validate the ENR-induced DNA methylation changes detected by RRBS, we performed targeted bisulfite sequencing of the activity-dependent transcription factor *Npas4*, which is known to mediate ENR-dependent hippocampal plasticity[50]. We confirmed the ENR-induced hypomethylation at several CpGs in the promoter region of *Npas4* (Fig. 1h; $n = 12$). Moreover, DNA methylation changes were associated with increased levels of *Npas4* transcript in the dentate gyrus of ENR compared to STD mice (Fig. 1i), suggesting a potential relationship between ENR-induced DNA methylation and gene expression changes.

**Age-related DNA methylation changes in the mouse dentate gyrus.** To investigate the influence of ENR on age-related DNA methylation changes in the dentate gyrus, RRBS was performed on dentate gyrus tissue from young (6.5-week-old) and aged (14-month-old) mice, which had lived in STD or ENR for 4 days (young) or over a year (aged). Previous studies showed that 14-month-old mice exhibit age-related behavioral deficits, including reduced locomotor activity, increased anxiety-like behavior, and decreased spatial memory[51], as well as changes in brain synapse composition that relate to reduced neuron function[52].

To determine age-related DNA methylation changes in the dentate gyrus, we compared DNA methylation profiles between young STD

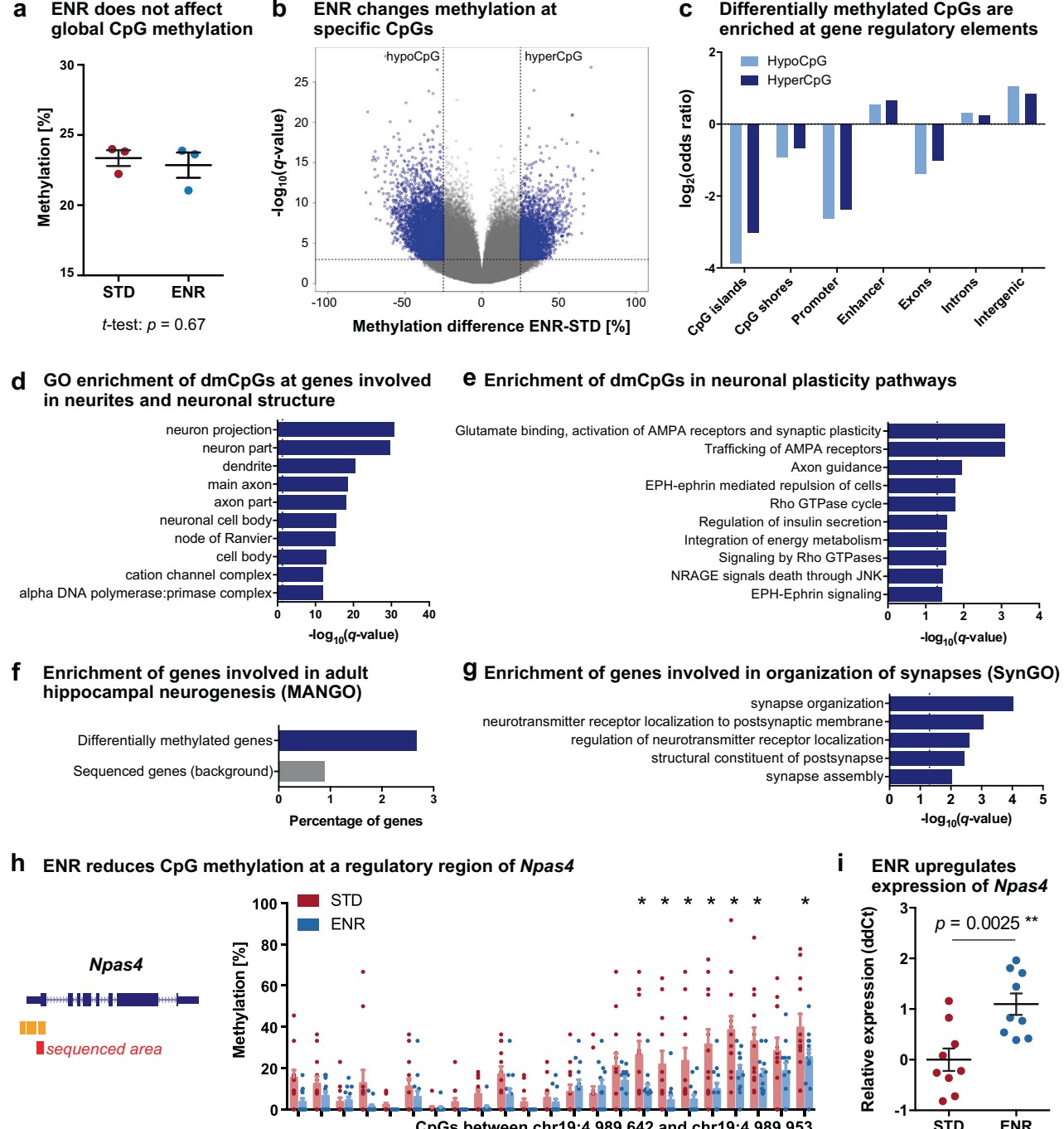

**a** ENR does not affect global CpG methylation

**b** ENR changes methylation at specific CpGs

**c** Differentially methylated CpGs are enriched at gene regulatory elements

**d** GO enrichment of dmCpGs at genes involved in neurites and neuronal structure

**e** Enrichment of dmCpGs in neuronal plasticity pathways

**f** Enrichment of genes involved in adult hippocampal neurogenesis (MANGO)

**g** Enrichment of genes involved in organization of synapses (SynGO)

**h** ENR reduces CpG methylation at a regulatory region of Npas4

**i** ENR upregulates expression of Npas4

mice and aged STD mice and detected 41,961 dmCpGs and 6600 dmCpHs (Supplementary Fig. 3a–c; Supplementary Data 3). While most dmCpGs were hypomethylated in aged mice (77.93% of dmCpGs), aging predominantly increased methylation of CpHs (74.05% of dmCpHs). The gene locations of aging-induced methylation changes in the dentate gyrus were significantly enriched with genes previously reported to exhibit age-related methylation changes in different tissues (Supplementary Fig. 3d). In addition, more than 92.0% of the here identified age-related differentially methylated genes have been shown to change methylation during aging in the mouse hippocampus by two independent studies[36,53] (Supplementary Fig. 3d), highlighting the robustness of aging-associated DNA methylation changes in the brain.

We found that age-related differentially methylated genes were significantly enriched in pathways related to neuronal plasticity,

neuronal signaling, and energy metabolism (Supplementary Fig. 3e; Supplementary Data 4). Remarkably, enriched pathways of genes with age-related DNA methylation changes overlapped significantly with enriched pathways of ENR-induced differentially methylated genes in the non-aged brain (hypergeometric test: $p = 2.1 \times 10^{-12}$), which suggested that aging and ENR changed DNA methylation at genes involved in similar pathways.

**ENR counteracts age-related DNA methylation changes**. To investigate the effect of life-long ENR on aging, we first compared global aging-induced DNA methylation changes between STD and ENR mice. In STD mice, aging was associated with a global 13.81% decrease in CpG methylation (Fig. 2a), which is in accordance with the predominant CpG hypomethylation found at

**Fig. 1 Environmental enrichment changes CpG methylation in the adult dentate gyrus at genes with a known role in neuronal plasticity, hippocampal neurogenesis, and synapse organization.** RRBS was performed on the dentate gyrus of 4.5-month-old mice housed in environmental enrichment (ENR) for three months and age-matched standard housed mice (STD; $n = 3$ individual mice per group). **a** ENR does not change global CpG methylation. Presented are data points for each mouse with group means ± standard errors of the mean (SEM); *p*-value is from two-sided, unpaired *t*-test. **b** Volcano plot highlighting ENR-induced differentially methylated CpGs (dmCpGs; methylation difference >25% and $q < 0.001$; linear regression with multiple testing correction using SLIM, see "Methods") in blue. Among the dmCpGs, 67.6% decreased methylation (hypoCpG) and 32.4% increased methylation in ENR (hyperCpG). **c** Genomic distribution of dmCpGs. HypoCpG and hyperCpG are depleted at CpG islands, CpG island shores, promoters, and exons ($\log_2$(odds ratio) < 0) and enriched at enhancers, introns, and intergenic regions ($\log_2$(odds ratio) > 0). Adjusted *p*-values < 0.001 for all regions (one-sided linear regression with multiple testing correction using false discovery rate; FDR). **d** Top ten significantly enriched cellular components from gene ontology (GO) enrichment analysis with differentially methylated genes (373 genes). Genes with at least four differentially methylated cytosines were used for enrichment analyses. **e** Top ten significantly enriched pathways of ENR-induced differentially methylated genes from the Reactome database. **f** Genes functionally involved in hippocampal neurogenesis (as annotated in the MANGO database) are enriched among ENR-induced differentially methylated genes (one-sided hypergeometric test: $p = 0.0020$). **g** Differentially methylated genes are involved in synapse organization and signaling. Depicted are significantly enriched biological processes from the SynGO database. The term "regulation of neurotransmitter receptor localization" was shortened from "regulation of neurotransmitter receptor localization to postsynaptic specialization membrane". **h** Validation of ENR-induced CpG methylation changes at *Npas4* using targeted bisulfite sequencing of the dentate gyrus from 12 individual mice per group (right panel). Methylation levels were compared at individual CpGs using repeated two-sided, unpaired *t*-tests with FDR correction (*adjusted $p < 0.05$). Depicted are individual data points for each mouse and bars with means + SEM. The left panel shows the location of the sequenced area (red bar) relative to *Npas4* exons and regulatory regions annotated by ENCODE (orange bars). **i** Quantitative polymerase chain reaction showed increased expression of *Npas4* in the dentate gyrus of ENR compared to STD mice (*p*-value from two-sided, unpaired *t*-test; $n = 9$ mice per group). Depicted are individual data points for each mouse and means ± SEM. Dashed lines in (**d**)–(**e**), (**g**) indicate significance thresholds of $q < 0.05$.

the individual CpG level (compare Supplementary Fig. 3). In contrast, aged ENR mice did not show significant global CpG methylation differences compared to young STD mice, suggesting that the age-related global CpG hypomethylation in the dentate gyrus is at least partially prevented by ENR. No ENR- and age-related global methylation changes were seen in the CpH context (Supplementary Fig. 4).

To analyze whether ENR counteracts aging-induced DNA methylation changes at specific genomic loci, we determined cytosines where the methylation change induced by lifelong ENR (difference between aged ENR and aged STD mice) was opposite to the effect of aging (difference between aged STD and young STD mice). From all CpGs that were hypomethylated with aging, 31.60% were significantly hypermethylated in aged ENR mice compared to aged STD mice (Fig. 2b). Similarly, from all CpGs hypermethylated with aging, 32.17% were hypomethylated by ENR. In the CpH context, 62.86% of aging-induced hypermethylated CpHs and 7.24% of hypomethylated CpHs were changed by ENR in the opposite direction than by aging (Fig. 2b). In total, 31.73% of all age-related dmCpGs and 48.42% of all aging-induced dmCpHs were counteracted by ENR. Among those cytosines, the vast majority (85.81% of CpGs; 98.47% of CpHs) did not show differences between young and aged ENR mice, confirming that, in ENR mice, methylation at these sites does not change with aging.

To compare magnitudes of methylation changes induced by aging and ENR, the absolute methylation percentages of the 13,314 dmCpGs and 3196 dmCpHs at which ENR counteracted aging effects were plotted (Fig. 2c–d). Compared to the young animals, the aged STD mice exhibited a loss of highly methylated CpGs, which resulted in an accumulation of low or unmethylated CpGs and 27% lower median CpG methylation levels. In contrast, the animals housed in ENR for one year showed distributions and median CpG methylation percentages similar to young animals. Likewise, aged STD mice possessed different distributions and a 33% increase in the median CpH methylation percentage compared to young animals (Fig. 2d–e), while aged ENR mice were similar to young animals. These results suggest that ENR restores methylation at that age-sensitive CpGs and CpHs to the levels observed in young animals.

To control that the detected overlaps between ENR-induced and age-related DNA methylation changes were not a result of chance, we randomly sampled 33,039 CpGs (referring to the number of ENR-

induced CpG methylation changes) from all sequenced CpGs (762,182) and analyzed their overlap with age-related CpG methylation changes (Supplementary Fig. 5a). On average 4.34% of randomly sampled CpGs overlapped with age-related differentially methylated CpGs, which was below the overlap of 31.73% detected for ENR-induced CpG methylation changes.

To test whether the detected overlap was a result of noise in the aged STD group, we randomly split the group of aged STD mice into two subsets and used one subset of three mice for the discovery of age-related DNA methylation changes and the other subset of three mice for calculating ENR-induced DNA methylation changes between aged STD and aged ENR mice (Supplementary Fig. 5b–c). Despite the reduced power due to smaller sample sizes, we detected significant overlaps between age-related and ENR-induced CpG methylation changes (Supplementary Fig. 5c), suggesting that the overlaps are not solely driven by noise. Moreover, we also found a significant overlap between aging and ENR effects when analyzing differentially methylated regions instead of individual CpGs using a different statistical approach (Supplementary Fig. 5d; see "Methods" for details).

To increase the temporal resolution of age-related DNA methylation changes, we integrated the data from the independent first experiment with 4.5-month-old (hereafter referred to as middle-aged) mice housed in ENR or STD for three months (compare Fig. 1). The CpG and CpH methylation levels of middle-aged mice were similar to young animals and did not show the age-related loss of CpG methylation observed in aged STD mice (Supplementary Fig. 6a). Further comparison of the effects of life-long ENR with age-related changes that occurred from middle-aged STD to aged STD mice showed that the pattern of dmCpGs and dmCpHs that were counteracted by ENR was similar to that observed for young animals (Supplementary Fig. 6b). In addition, 43.71% of CpGs and 56.32% of CpHs at which ENR counteracted aging were also differentially methylated between middle-aged STD mice and aged STD mice. This suggests that age-related epigenetic changes became pronounced only after an age of 4.5 months and that the environmental influence on age-related methylation changes is itself age-dependent.

**Genomic distribution of ENR- and age-dependent DNA methylation.** To further characterize the influence of ENR on aging-induced epigenetic reconfigurations, we analyzed the

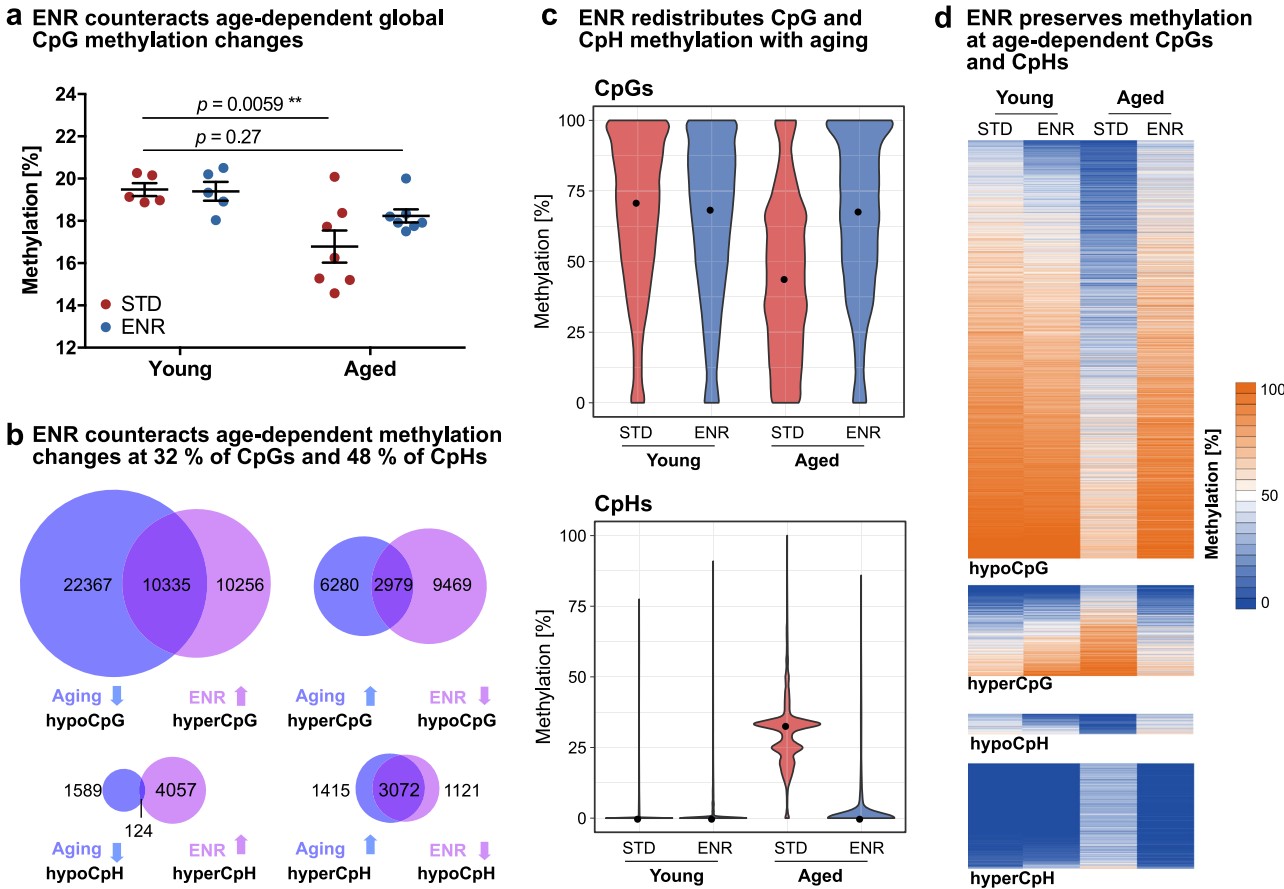

**Fig. 2 Environmental enrichment protects the dentate gyrus from age-related CpG and CpH methylation changes.** DNA methylation profiles of young mice housed in ENR or STD for four days (Young ENR; Young STD; $n = 5$) and aged mice housed in ENR or STD for one year (aged ENR; Aged STD; $n = 7$) as determined by RRBS. **a** Aging decreases global CpG methylation in STD but not in ENR mice ($p$-values from two-way ANOVA with Dunnett's post hoc test; depicted are means ± SEM). **b** Aging and ENR induce opposite DNA methylation changes. Overlap of aging-induced hypoCpG with ENR-induced hyperCpG in aged mice (top left) and aging-induced hyperCpGs with ENR-induced hypoCpGs (top right). Overlap of aging- and ENR-induced CpH methylation changes (bottom). Significant overlaps with $p < 1 \times 10^{-100}$ for all comparisons as determined by one-sided hypergeometric tests. **c** Age- and ENR-dependent changes of the distributions of methylation percentages of the 13,314 CpGs and 3,196 CpHs counteracted by ENR. Medians of methylation percentages per group are highlighted as black dots (CpGs; Young STD: 71.05%, Young ENR: 68.57%, Aged STD: 44.03%, Aged ENR: 67.93%. CpHs; Young-STD: 0%, Young-ENR: 0%, Aged-STD: 32.85%, Aged ENR: 0%). **d** Heatmaps depicting the absolute methylation percentages in the individual groups at individual CpGs and CpHs. While aged STD mice show distinct methylation at individual sites, aged ENR mice show methylation levels similar to young animals.

genomic distribution of CpGs and CpHs at which ENR counteracted age-related methylation changes. Both dmCpGs and dmCpHs were distributed over all chromosomes in the genome (Fig. 3a), with dmCpGs depleted at exons, CpG islands, CpG island shores, promoters, and super-enhancers, but significantly enriched at introns, intergenic regions, and enhancers (Fig. 3b). The distribution of dmCpHs over genomic features was similar to dmCpGs, but they were not enriched at enhancers. Because DNA methylation of enhancers is particularly involved in the regulation of gene expression by interaction with transcription factors[54], we performed a transcription factor motif analysis of the 1472 dmCpGs located within enhancers. The only significantly enriched transcription factor was Mecp2 (Fig. 3c), which binds methylated cytosines via its methyl-CpG-binding domain[55]. Mecp2 was also the strongest enriched transcription factor when motif enrichment analysis was performed on all 13,314 dmCpGs (independent of genomic location), which suggested that Mecp2 binding occurs genome-wide and is not restricted to dmCpGs located within enhancers (Supplementary Data 5). In contrast to dmCpGs, no enrichment of Mecp2 motifs was found at dmCpHs (Fig. 3c).

To validate the enrichment of Mecp2 binding sites at dmCpGs, we used two existing Mecp2 ChIP-sequencing datasets derived from adult mouse cortex and prefrontal cortex, respectively[56]. Similar to Mecp2 motifs, ChIP-derived Mecp2-bound genomic regions were significantly enriched at dmCpGs (hypergeometric tests; cortex: $p < 1 \times 10^{-300}$; prefrontal cortex: $p = 1.4 \times 10^{-286}$).

Because Mecp2 is known to specifically bind methylated cytosines, we analyzed the direction of the methylation change of Mecp2 target dmCpGs with aging and ENR. The vast majority of Mecp2 targets (87.31%) were hypomethylated with aging and hypermethylated by ENR (Fig. 3d). Aging-induced hypomethylated CpGs were significantly enriched among the Mecp2 targets ($p = 2.7 \times 10^{-36}$). These results suggest that ENR prevents the age-associated CpG hypomethylation of Mecp2 binding sites and, thereby, potentially facilitates Mecp2 binding in the aged dentate gyrus.

Annotation of dmCpGs and dmCpHs to their associated gene targets identified 676 genes at which ENR counteracted age-related methylation changes. Two-thirds of these genes were protein-coding genes, but also lncRNA and a minor percentage of ncRNAs (such as microRNAs) and pseudogenes changed DNA

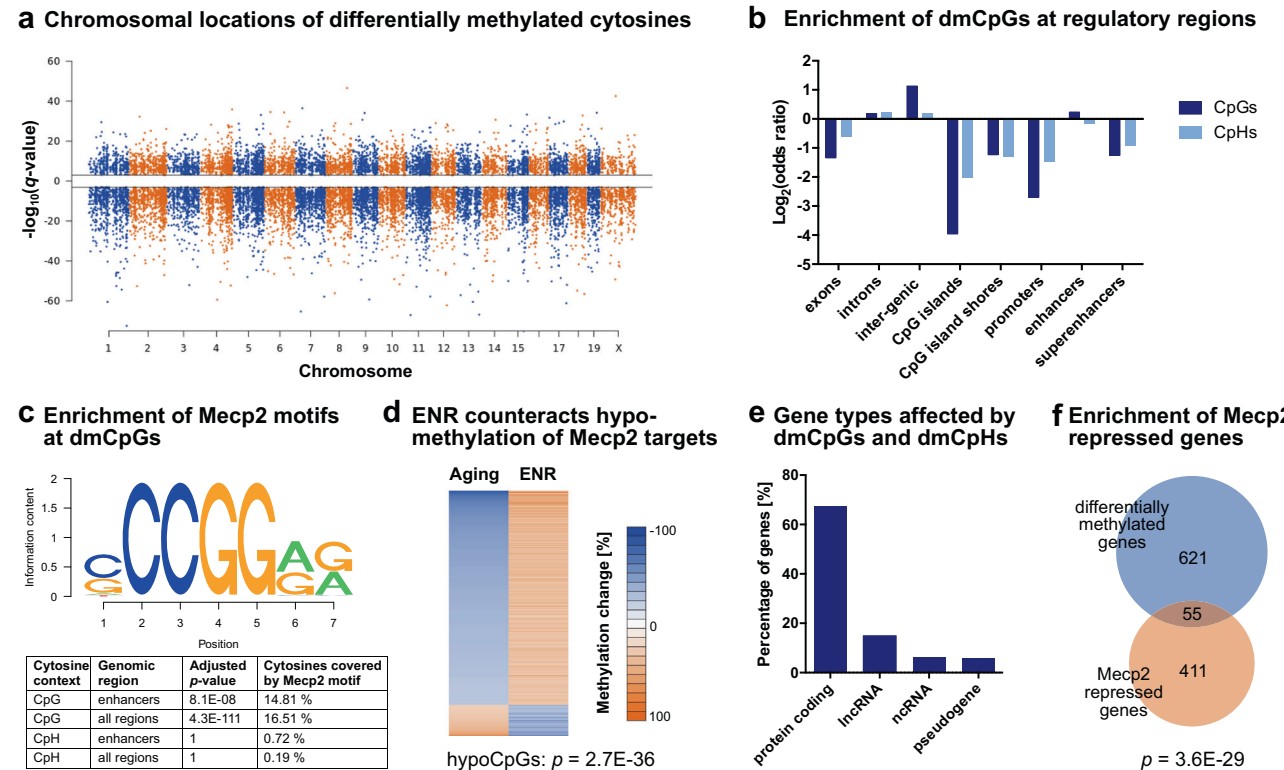

**Fig. 3 Genomic distribution of cytosines at which environmental enrichment counteracts age-related methylation changes. a** Manhattan plot depicting the distribution of dmCpGs and dmCpHs over chromosomes. Q-values indicate the significance of the aging-induced methylation changes with the sign corresponding to the direction of the change (negative $-\log_{10}$(q-value) for hypomethylated cytosines; positive $-\log_{10}$(q-value) for hypermethylated cytosines). **b** CpGs and CpHs are depleted at CpG islands, CpG island shores, promoters, and exons ($\log_2$(odds ratio) < 0), but enriched at introns and intergenic regions of the genome ($\log_2$(odds ratio) > 0). In addition, CpGs but not CpHs are enriched at enhancer regions. Adjusted $p$ < 0.001 for all regions, except CpHs at enhancers (adjusted $p$ = 0.061; one-sided linear regression with FDR correction). **c** Transcription factor binding analysis revealed enrichment of Mecp2 motifs at dmCpGs located within enhancers (1472 CpGs) as well as at all regulated 13,314 dmCpGs (independent of genomic location). Mecp2 motifs were not enriched at dmCpHs. Depicted are the Mecp2 motif as annotated in the database motifDb and the results of the enrichment analysis (one-sided hypergeometric test with FDR correction). **d** The 2198 dmCpGs that fall into Mecp2 targets sites are enriched with CpGs that were hypomethylated with aging ($p$-value from one-sided hypergeometric test compared to all 13,314 CpGs). **e** Classification of gene targets of cytosines at which ENR counteracts aging-induced methylation changes (676 differentially methylated genes). **f** Previously identified Mecp2-repressed genes[68] are enriched among the differentially methylated genes ($p$-value from the one-sided hypergeometric test).

methylation (Fig. 3e). The differentially methylated genes were significantly enriched with genes that are known to be transcriptionally repressed by Mecp2[56] (Fig. 3f), further supporting that ENR counteracts age-related DNA methylation changes at Mecp2 targets.

**ENR preserves brain plasticity genes in the aged dentate gyrus.** To identify aspects of brain aging that are sensitive to environmental stimulation, we performed GO and pathway enrichment analyses of genes at which ENR counteracted aging-induced methylation changes and generated an integrated map based on gene overlaps between enriched terms[57]. We identified three main clusters which were related to neuronal plasticity, cell communication, and developmental programs (Fig. 4a; Supplementary Data 6). Among those, the "neuron plasticity" cluster contained the highest number of enriched terms. It comprised pathways related to structural plasticity of neurons, including axon guidance and synapse organization, as well as processes involved in neuronal activation, such as synaptic signaling and regulation of potassium channels. In the "cell communication" cluster, we found signaling pathways that involve secreted molecules, such as insulin and growth factors, and a group of enriched terms related to cell adhesion, including the formation

of cellular junctions by cadherins, adhesion to the extracellular matrix, and extracellular matrix remodeling by regulation of collagen formation. The third cluster contained pathways involved in organ morphogenesis but also pathways related to nervous system development, including neuronal fate specification and precursor proliferation.

To correlate differential cytosine methylation with cellular changes in the brain, we investigated adult hippocampal neurogenesis which has a role in cognitive flexibility and decreases with aging[20]. We found that genes at which ENR counteracted age-related methylation changes were significantly enriched for genes functionally involved in adult hippocampal neurogenesis as annotated in MANGO (Fig. 4b). Accordingly, mice housed in ENR for one year showed a 60% increase in the number of newborn neurons in the dentate gyrus compared to aged STD mice (Fig. 4c–d). We also observed elevated ENR-stimulated neurogenesis in the non-aged brain (Fig. 4c), together suggesting that continuous ENR increases hippocampal neurogenesis throughout the lifespan. In contrast, only trends toward increased precursor proliferation and no change in total precursor cell numbers were observed in aged ENR compared to aged STD mice (Supplementary Fig. 7). This suggests that life-long ENR stimulates newborn neurons during an immature phase but has only minor effects on neural precursor cells, which is in

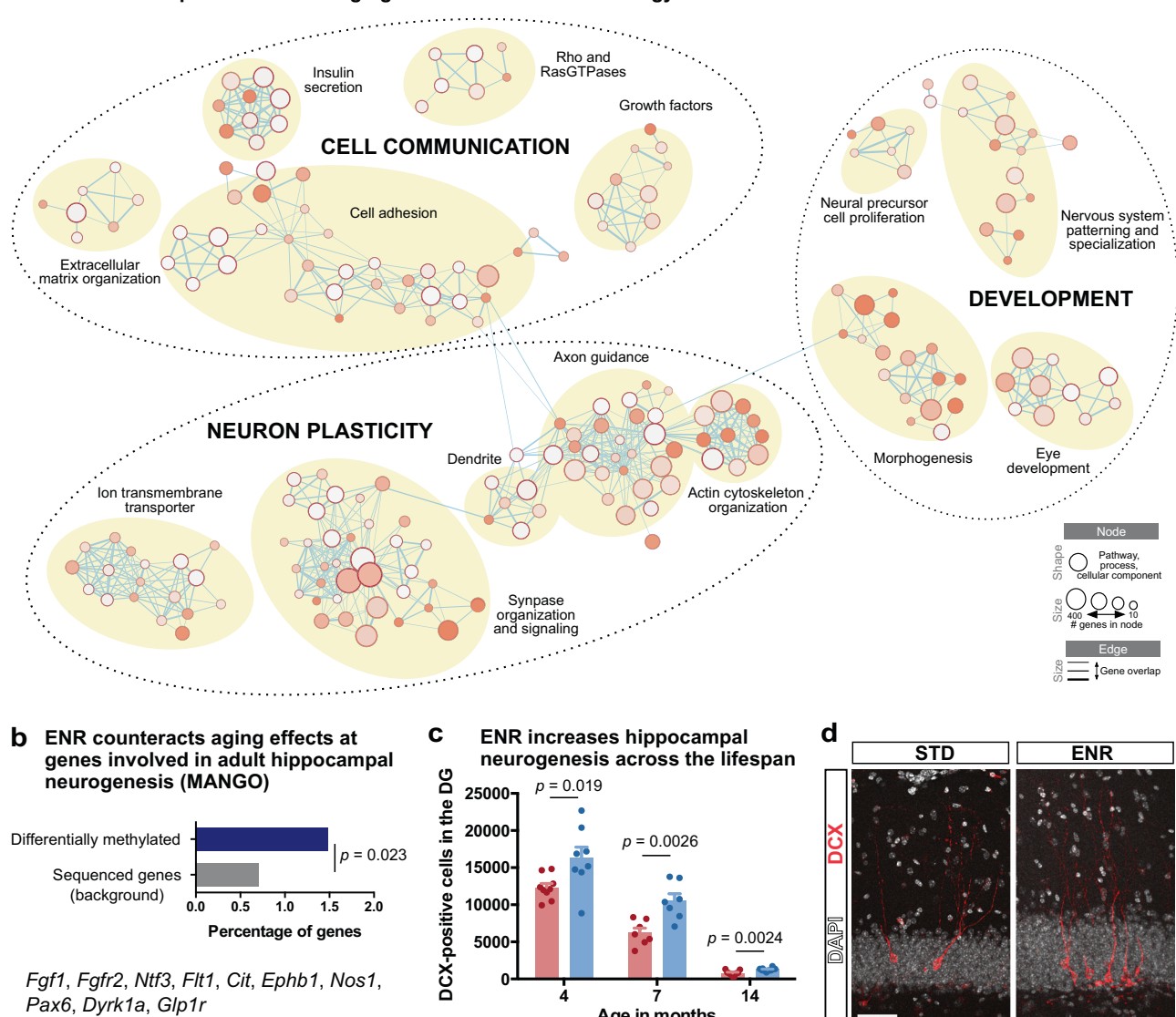

**a** Functional map of ENR versus aging interactions in the dentate gyrus

**b** ENR counteracts aging effects at genes involved in adult hippocampal neurogenesis (MANGO)

*Fgf1, Fgfr2, Ntf3, Flt1, Cit, Ephb1, Nos1, Pax6, Dyrk1a, Glp1r*

**c** ENR increases hippocampal neurogenesis across the lifespan

**d** STD   ENR

**Fig. 4 Functional enrichment connects environmentally sensitive age-related DNA methylation changes with neuronal plasticity and adult hippocampal neurogenesis. a** Integrated map of GO terms, SynGO terms, Reactome pathways, and "WikiPathway" terms enriched among the 676 ENR-induced differentially methylated genes that counteract aging effects (significantly enriched terms with $q < 0.05$). Three main functional clusters were identified (dashed circles), each containing several groups of highly connected terms (yellow circles). Legend indicates node and edge sizes. **b** Genes functionally involved in hippocampal neurogenesis are enriched among differentially methylated genes at which ENR counteracts age effects (p-value from a one-sided hypergeometric test). **c** Mice housed in ENR starting at an age of 6 weeks have significantly more newborn, doublecortin (DCX)-expressing neurons in the dentate gyrus (DG) at ages 4 months, 7 months, and 14 months compared to STD housed mice. Depicted are individual data points and bars with group means + SEM (4 months: $n = 9$ STD mice, 8 ENR mice; 7 months: $n = 7$ mice per group; 14 months: $n = 9$ STD mice, 10 ENR mice). Depicted p-values are from a two-sided, unpaired t-test. **d** Representative image of fluorescent staining to detect newborn neurons in STD and ENR mice. Fluorescent stainings were reproduced for all samples presented in panel (**c**). Scale bar: 50 μm.

accordance with what has been observed after shorter periods of ENR in young adult and aged mice[20].

Genes containing age-related differentially methylated cytosines that were not influenced by ENR showed a similar enrichment at neuronal plasticity pathways as the genes counteracted by ENR (Supplementary Fig. 8a). Indeed, 341 of the genes at which ENR protected from age-related DNA methylation changes did also contain age-related differentially methylated cytosines that were not influenced by ENR. Moreover, the cytosines at which ENR did not significantly alter age-related DNA methylation changes showed a similar genomic distribution

as the loci at which ENR counteracted aging effects (Supplementary Fig. 8b). These results suggest that ENR does not affect a specific subset of pathways underlying brain aging, but rather partially rescues global aging effects in the brain.

To analyze whether ENR-induced changes at diverse genomic regions might differ in function, Reactome pathway and MANGO enrichment analyses were performed separately for genes with DNA methylation changes in enhancers, promoters, or gene bodies (Supplementary Data 7). We found a significant enrichment of neuronal plasticity and neurogenesis pathways specifically for ENR-induced gene body methylation. However, we also

found neuronal plasticity-related genes with ENR-induced changes at enhancers such as *Vegfa* and *Flt1*. No differences in functional annotation were detected between hypo- and hyper-methylated genes (Supplementary Data 7).

To analyze whether ENR protects from aging effects by actively changing DNA methylation, we overlapped genes where ENR counteracted DNA methylation changes in aged mice with DNA methylation changes induced by shorter periods of ENR in non-aged mice. In total, 24.02% of genes that changed DNA methylation after four days of ENR in young mice and 30.03% of genes that changed DNA methylation after three months of ENR in middle-aged mice overlapped with genes at which ENR counteracted age-related changes (Supplementary Fig. 9a). In addition, ENR-induced differentially methylated genes were significantly enriched with genes that changed transcription after acute neuronal activation in the dentate gyrus[58] (Supplementary Fig. 9b). These results might suggest that active ENR-induced DNA methylation changes at neuronal activity-regulated genes counteract the development of age-related DNA methylation changes.

**ENR housing of aged mice compensates age-related DNA methylation**. Our results so far suggested that lifelong ENR prevented the development of age-related DNA methylation changes, but it remained unknown whether ENR could reverse existing age-related DNA methylation changes in old mice. To test the latter, we housed 14-month-old mice in ENR or STD for three months. We chose to start ENR housing at an age of 14 months because this was the age at which we had confirmed the existence of age-related DNA methylation changes in our previous experiment (Supplementary Fig. 3). To avoid a potential influence of ENR on postnatal brain development, we used 3-month-old mice as the young control group.

We first calculated DNA methylation changes between the 3-month-old and 17-month-old mice and confirmed 65.6% of the age-related differentially methylated genes identified in our previous experiment (hypergeometric test: $p < 1 \times 10^{-300}$; Supplementary Fig. 10a). When we overlapped age-associated changes with ENR-induced methylation changes in aged mice, we found that ENR of aged mice counteracted aging effects at 36.52% of CpGs and 30.33% of CpHs (Supplementary Fig. 10b–d). The associated gene loci significantly overlapped with the genes influenced by lifelong ENR (hypergeometric test: $p = 2.2 \times 10^{-186}$; Fig. 5a) and showed a similar enrichment at genes involved in synaptic plasticity and adult hippocampal neurogenesis (Fig. 5b). Moreover, the pattern of the DNA methylation changes (Supplementary Fig. 10b–d), the genomic distribution of the affected cytosines (Supplementary Fig. 10e), and the significant enrichment at Mecp2 binding sites was similar to what we had observed after lifelong ENR (Supplementary Fig. 10f). Together, these results confirmed our previous findings and suggested that ENR can restore young DNA methylation patterns in the dentate gyrus of old mice.

To further explore potential mechanisms of how ENR-induced DNA methylation changes could improve brain function during aging, we examined the influence of ENR on Mecp2 binding in the aged dentate gyrus. Mecp2 is a well-known regulator of neuron function which is critical for embryonic and postnatal brain development[59] but had not been linked to brain aging yet. First, we selected candidate loci at which our RRBS data showed that ENR counteracted age-related DNA methylation changes and which contained Mecp2 motifs as determined by our transcription factor binding analyses (Fig. 5c–d). Specifically, we chose a locus within an intragenic enhancer near the gene *Tiam1*, which is known to control hippocampal plasticity[60], as well as an intragenic enhancer near the gene *Tshz2*, a transcriptional

repressor that had not been linked to brain aging before. We then validated the aging- and ENR-induced DNA methylation changes at those loci using targeted bisulfite sequencing (Fig. 5e). To investigate potential changes in Mecp2 binding, we performed chromatin immunoprecipitation with a validated antibody against Mecp2[61] followed by quantitative PCR of the genomic area surrounding the Mecp2 motifs. For the chromatin preparation, we used dentate gyri from the opposite hemisphere of the same mice that were used for DNA methylation profiling (17-month-old mice housed in ENR or STD for three months; young 12-week-old mice). We detected significantly reduced binding of Mecp2 in aged STD mice compared to young STD mice at the indicated loci within the *Tiam1* gene (Fig. 5f). In contrast, aged ENR mice showed increased levels of Mecp2 binding compared to aged STD mice. Mecp2 has been shown to act as a transcriptional repressor, although roles in locus-specific gene activation and regulation of alternative splicing have been reported[62]. To analyze potential consequences of Mecp2 binding on transcription, we performed quantitative RT-PCR of *Tiam1* using dentate gyrus tissue from an independent cohort of 12-week-old young mice and 17-month-old mice housed in ENR or STD for three months and found a small age-related increase in expression, which was counteracted by ENR (Fig. 5g). Similar results were found for the analyzed locus at *Tshz2* (Fig. 5h–k). In addition to candidates *Tiam1* and *Tshz2*, we investigated the neuronal transcription factor *Cux2*[63], which contained an intra-genic Mecp2 motif with an age-related loss of DNA methylation that was counteracted by lifelong ENR (Fig. 5l) and by three months ENR of aged mice (Fig. 5m). The DNA methylation changes at *Cux2* correlated with age- and ENR-induced changes in Mecp2 binding (Fig. 5n), while no significant difference in the levels of *Cux2* transcript was observed between the groups (Fig. 5o), suggesting that effects of altered Mecp2 binding on gene expression in the aged brain are locus-specific. Accordingly, when we overlapped ENR-induced DNA methylation changes with previously published RNA sequencing data of the dentate gyrus of ENR housed mice[64], we found a significant enrichment of ENR-induced hypermethylation among both genes up-regulated and down-regulated after ENR (Supplementary Fig. 11). Together, these results indicated that ENR housing of aged mice counteracts the age-related loss of DNA methylation at Mecp2 motifs and counteracts associated changes in Mecp2 binding and gene expression at selected candidate loci.

**ENR influences genes with known roles in human cognitive decline**. To investigate whether the genes at which long-term ENR counteracted age-related DNA methylation changes in the mouse brain are also dysregulated in the human brain during aging and associated cognitive decline, we overlapped them with three previously published datasets derived from human pre-frontal cortex tissue. Dataset 1 contained genes with DNA methylation changes associated with Alzheimer's disease pathology[38]; datasets 2 comprised genes with changes in RNA levels associated with age-related cognitive decline[65] and dataset 3 contained genes with proteomic changes associated with individual cognitive trajectories during aging[66]. Genes of all three datasets were significantly enriched among the genes at which one year of ENR counteracted age-related DNA methylation changes (Fig. 6a). In total, 219 of the ENR-induced differentially methylated genes detected in the aged mouse brain were also dysregulated in the aged human brain (Fig. 6b; Supplementary Data 8). Among those genes, the most significantly enriched biological process from GO analysis was the term "neurogenesis" (adjusted $p = 8.3 \times 10^{-08}$; 53 genes) of the higher-order group "nervous system development" (adjusted $p = 1.3 \times 10^{-09}$; 71

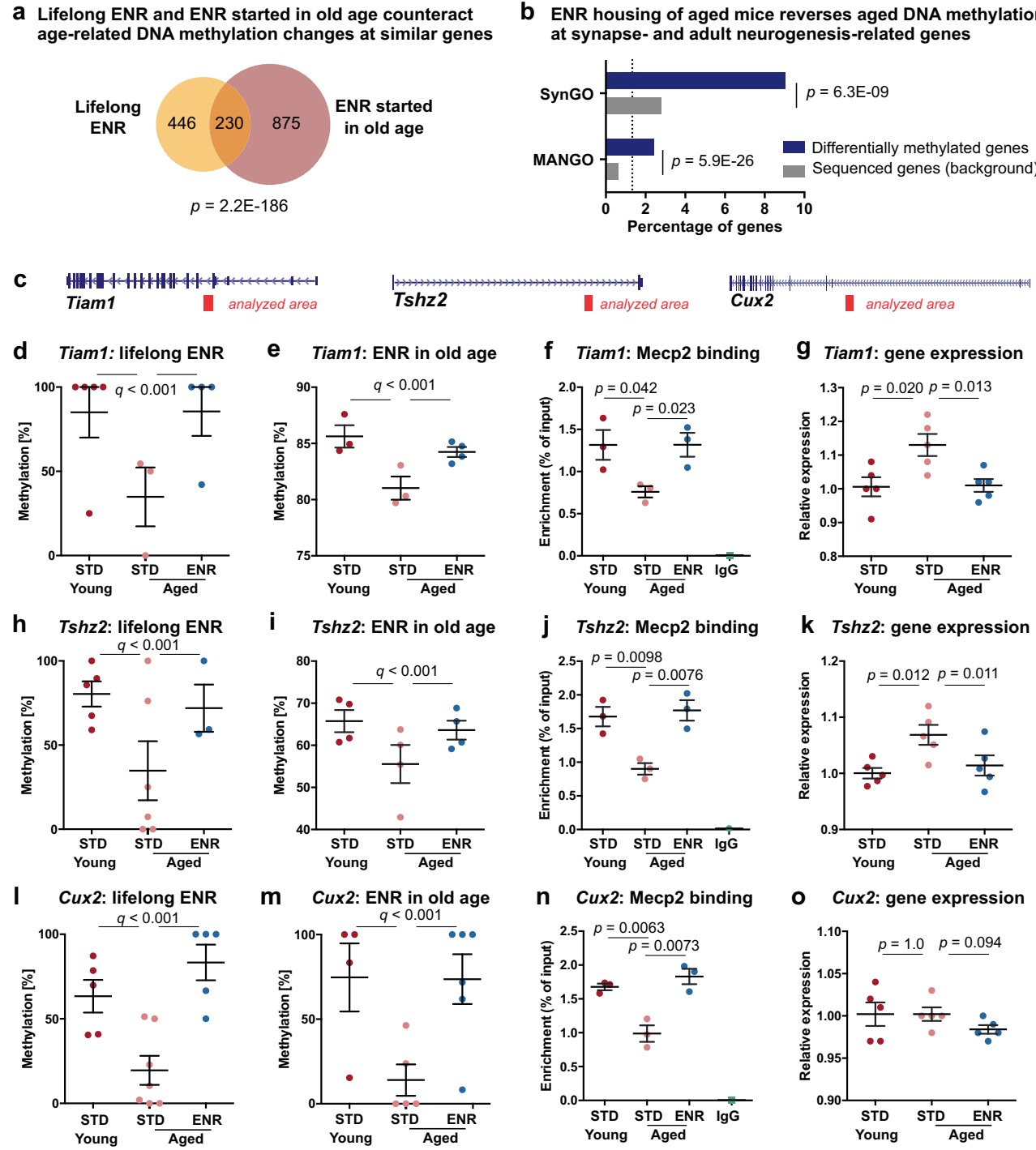

**a** Lifelong ENR and ENR started in old age counteract age-related DNA methylation changes at similar genes

**b** ENR housing of aged mice reverses aged DNA methylation at synapse- and adult neurogenesis-related genes

**d** *Tiam1:* lifelong ENR

**e** *Tiam1*: ENR in old age

**f** *Tiam1*: Mecp2 binding

**g** *Tiam1*: gene expression

**h** *Tshz2*: lifelong ENR

**i** *Tshz2*: ENR in old age

**j** *Tshz2*: Mecp2 binding

**k** *Tshz2*: gene expression

**l** *Cux2*: lifelong ENR

**m** *Cux2*: ENR in old age

**n** *Cux2*: Mecp2 binding

**o** *Cux2*: gene expression

genes). This suggests that the age-related dysregulation of genes involved in neurogenesis is conserved between mouse and humans. The overlap of genes regulated by ENR in mice and genes dysregulated in human brains highlights the intriguing possibility that environmental stimulation could prevent age-related gene dysregulation and associated cognitive decline also in human brains.

## Discussion

Aging-induced epigenetic changes are conserved among species[67] and contribute to age-related brain dysfunctions, including reductions in cognition and hippocampal neurogenesis[27,28]. Behavioral interventions and environmental stimulation are known to alter epigenetic modifications in the brain[33,64,68], but their influence on age-related epigenetic changes was unclear. Here, we showed that lifelong ENR can counteract age-related DNA methylation changes in the hippocampal dentate gyrus. Many of the genes for which ENR prevented aging-induced differential methylation in the mouse brain were also known to be dysregulated in human brains with aging and related cognitive decline. Our results lend mechanistic support to the potential of behavioral interventions and lifestyle factors to prevent age-related epigenetic changes in the brain in order to improve brain health in old age.

We found that ENR prevented the aging-induced loss of CpG methylation at motifs of Mecp2, which binds methylated

**Fig. 5 Environmental enrichment compensates changes in DNA methylation, Mecp2 binding, and transcription at neuronal plasticity genes. a** Overlap of genes at which ENR counteracted age-related DNA methylation changes after one year of ENR (lifelong) and three months of ENR in aged mice. Venn diagram considered the direction of the DNA methylation change. **b** Enrichment of gene loci at synapse-related genes (SynGO) and at adult neurogenesis-related genes (MANGO). **c** Images of the genes *Tiam1*, *Tshz2*, and *Cux2* highlighting the analyzed gene region in red. **d** RRBS profile of young, aged STD and aged ENR mice (one year) at *Tiam1* (chr16:89,896,498). Data correspond to the experiment depicted in Figs. 2–4. **e** Targeted bisulfite sequencing at chr16:89,896,498 of young, aged STD and aged ENR mice (3 months ENR; $n = 3$ STD mice, $n = 4$ ENR mice). **f** Relative Mecp2 binding at *Tiam1* in young, aged STD and aged ENR mice (three months ENR). **g** Relative expression of *Tiam1* increased with aging and was reduced in aged ENR mice (three months ENR; $n = 5$ mice per group). **h–i** Loss of DNA methylation at *Tshz2* locus (chr2:169,826,488) with aging and increased methylation with ENR as shown by RRBS (**h**) and targeted bisulfite sequencing (**i**) relates to changes in Mecp2 binding (**j**) and gene expression (**k**). Methylation changes at *Cux2* (chr5:121,938,836) as determined by RRBS (**l**)–(**m**) correlated with changes in Mecp2 binding (**n**) but did not affect transcription (**o**). Sample information: Sample sizes for (**d**), (**h**), (**l**) were $n = 5$ young mice and $n = 7$ aged mice per group (as indicated in Fig. 2). The sample size for panel m was $n = 8$ mice per group (as indicated in Supplementary Fig. 10). Depicted data points in (**d**), (**e**), (**h**), (**l**), (**m**) represent the methylation levels of all individual mice for which that CpG locus was covered by RRBS. Sample sizes for (**f**), (**j**), (**m**) were $n = 3$ samples per group (samples refer to pools of dentate gyri from three mice). Data in (**d**)–(**o**) are represented as individual data points for each mouse with group means ± SEM. Sample sizes for (**g**), (**k**), (**o**) were $n = 5$ mice per group. Depicted *q*-values in (**d**), (**e**), (**h**), (**i**), (**l**), (**m**) are from logistic regression (two-sided) with SLIM correction; *p*-values in (**a**) and (**b**) are from one-sided hypergeometric test; *p*-values in (**f**), (**g**), (**j**), (**k**), (**n**), (**o**) are from two-sided, unpaired *t*-test (no adjustment for multiple comparisons).

cytosines with high affinity broadly across the genome[69–71]. Moreover, our candidate-specific analysis showed that aging was associated with reduced Mecp2 binding to neuronal genomes and that ENR enhanced Mecp2 binding in aged brains. Mutations in Mecp2 cause Rett syndrome, which is characterized by severe encephalopathy and reduced lifespan[72]. Mecp2 is known to be abundant in neurons and crucial for neuronal development during embryogenesis and in the adult hippocampus[73,74], but the role of Mecp2 in brain aging had not yet been reported. At the molecular level, Mecp2 has been associated with transcriptional repression, locus-specific gene activation, and regulation of alternative splicing[70,71,75]. Therefore, aging-induced loss of Mecp2 binding as a result of genome-wide CpG hypomethylation might be involved in the aging-associated transcriptional dysregulation and the aberrant splicing of neuronal genes that have been reported for the aged hippocampus[76]. On the other hand, a recent study suggested that Mecp2 is involved in the repression of endogenous repetitive elements in neuronal genomes[77]. ENR-induced methylation of Mecp2 targets could thus prevent the aberrant activation of repetitive elements which is known to contribute to genomic instability in aging[78]. Follow-up studies should confirm the interaction between CpG hypomethylation and Mecp2 binding during brain aging on a genome-wide scale and identify molecular programs downstream of Mecp2 that are disrupted in the aged brain. Our results suggest not only a possible role of Mecp2 in age-related cognitive decline but also the potential of environmental stimulation to rescue Mecp2 binding in aged brains by DNA methylation of its target sites.

With the exception of a few regulated genes[79,80], the molecular mechanisms underlying ENR-stimulated brain plasticity were, until now, mostly unknown. Our functional enrichment analyses of differentially methylated genes in the aged mouse dentate gyrus have exposed several pathways potentially underlying the neuroprotective effects of ENR during aging. One such pathway was adult hippocampal neurogenesis and, since increasing neurogenesis promotes aspects of hippocampal function[81], ENR-stimulated increases in hippocampal neurogenesis likely ameliorate age-related cognitive decline. Further investigation will be needed to determine the cell type specificity of ENR-induced DNA methylation changes in the dentate gyrus and their role in hippocampal neurogenesis.

Since ENR increased adult hippocampal neurogenesis throughout the lifespan, differences in DNA methylation patterns between STD and ENR mice could reflect DNA methylation changes in mature neurons or ENR-induced differences in the cellular composition of the dentate gyrus. Based on our quantifications of newborn neuron numbers, supported by previous quantifications of total granule cell numbers in the dentate gyrus[82], the percentage of newborn neurons among all neurons increased from 0.21% in aged STD mice to 0.31% in aged ENR mice. Because our DNA methylation analysis only considered cytosines with absolute DNA methylation differences larger than 25% as dmCpGs or dmCpHs, DNA methylation differences as a result of an increased percentage of new neurons in ENR mice would have not been detected. Second, because hippocampal neurogenesis sharply decreases in young adulthood and plateaus thereafter[83], differences in DNA methylation patterns as a result of differences in newborn neuron numbers should have been visible in the 4.5-month-old "middle-aged" mice. However, the middle-aged mice showed only minor age-related DNA methylation changes compared to young animals (Supplementary Fig. 4). Therefore, we conclude that the here described ENR-induced DNA methylation changes that counteracted age effects reflect DNA methylation changes in mature hippocampal neurons rather than changes in the cellular composition of the dentate gyrus.

ENR combines physical and social incentives with continuous novelty and sensory stimulation[84]. Which aspects of ENR and the specific mechanisms of how environmental stimulation prevents age-related DNA methylation changes remain to be investigated. Previous work has shown that exploration of novel environments leads to neuronal activation coupled with short-term and long-lasting changes in transcription and chromatin accessibility in activated hippocampal neurons[85,86]. In addition, isolated neuronal activation by electroconvulsive stimulation has been shown to lead to widespread changes in DNA methylation, RNA, and open chromatin landscapes in the dentate gyrus[34,58]. We showed that ENR-induced differentially methylated genes were significantly enriched for neuronal activity-regulated genes, indicating that some of the ENR-induced DNA methylation changes are triggered by neuronal activation. Similarly, Penner et al. reported that age-related DNA methylation changes at the promoter of the neuronal activity-induced gene *Egr1* were reversed in the hippocampus by acute exploration of a novel environment[37]. We found that many of the genes at which ENR counteracted age-related DNA methylation changes were also differentially methylated by acute (4 days) ENR in non-aged brains or by 3 months ENR in middle-aged mice. Therefore, the compensating effects of ENR on age-related DNA methylation changes are potentially mediated by repeated neuronal activation through continuous novelty stimulation in ENR. Future studies should specify how ENR interacts with age-related DNA methylation changes at the molecular and cellular levels.

**a** Enrichment of genes dysregulated in human brain aging and cognitive decline

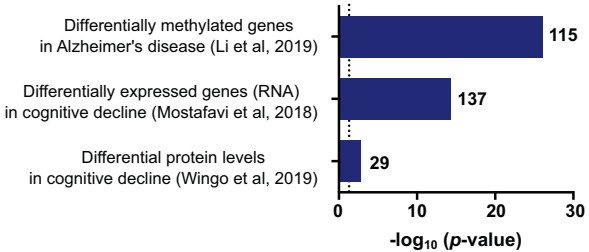

**b** Network of human brain aging genes counteracted by ENR in mice

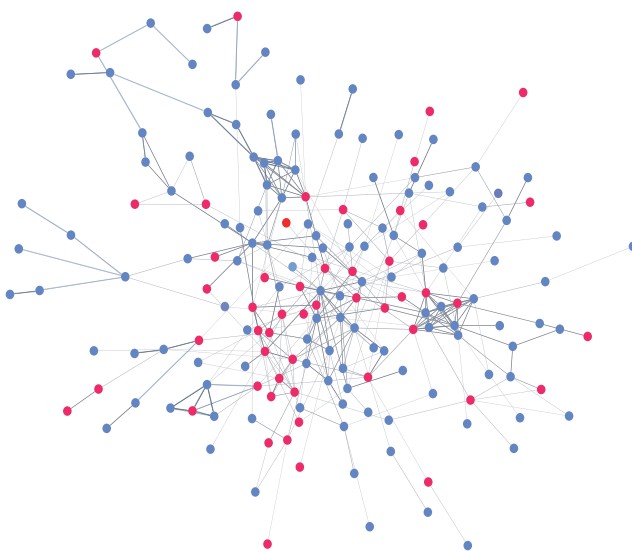

**Fig. 6 Genes at which ENR counteracts age-related methylation changes are dysregulated in the human brain in age-related cognitive decline.**
**a** Genes with DNA methylation changes[38], differential RNA[65], or protein[66] levels in the human prefrontal cortex associated with age-related cognitive decline are significantly enriched among the genes that are differentially methylated with aging and 1 year ENR in the mouse dentate gyrus. For overlap with differentially methylated genes[38], direction of the DNA methylation changes was taken into account. For differentially expressed genes and differential proteins, genes were overlapped independent of the direction of the age-related DNA methylation changes. Depicted are the $-\log_{10}$(p-values) from one-sided hypergeometric tests and the number of genes that overlapped between datasets. The dashed line indicates a significance threshold of $p = 0.05$. **b** STRING protein interaction network of human brain aging genes at which ENR counteracted age-related DNA methylation changes in the mouse brain. Genes involved in the highest enriched GO biological process "neurogenesis" are highlighted in red.

The paradigm of ENR is a highly reductionistic construct, which addresses relative phenotypical changes that can be ascribed to environmental, non-genetic effects. By using inbred mice the genetic factor is tightly controlled for. Exposure to ENR elicits a large number of effects on the brain and behavior and the idea of "environmental enrichment" has been translated to the human situation to describe the impact of environmental influences on brain development, cognition, and especially neurorehabilitation. Humans show substantial inter-individual differences in their behavioral activity and the ones with a physically and mentally active lifestyle and social engagements are more likely to maintain brain function and health in old age[1,2]. They also show

considerable genetic variability, however, so that the experimental ENR construct in animal studies is not identical to what can be observed and more loosely described as an "enriched environment" in humans. While this is a critically important caveat, particularly relevant for judging effect sizes, animal studies offer a perspective that is almost inaccessible in humans. Effectively, what the paradigm allows estimating is the impact of environment and activity, if there were no genetic variation. We are thus not making claims about precise equivalents of our findings under human conditions and not even under feral conditions of the animals but about patterns of change that might be conserved (albeit obscured) and principles of underlying mechanisms. In the present study, we found that ENR counteracted aging effects in mice at genes that are dysregulated with age-related cognitive decline in humans. Since, to our knowledge, no comparable molecular data exists yet for the human hippocampus, we compared genes dysregulated in the human prefrontal cortex with our epigenetic data from the mouse dentate gyrus. Whether ENR effects on DNA methylation patterns in aging are specific to the dentate gyrus is currently unknown. However, since ENR modifies neuronal activity in many brain areas, including the prefrontal cortex[87], the epigenetic effects are likely comparable between different brain areas. Furthermore, there is evidence that age-related DNA methylation changes are not tissue-specific, such that similar DNA methylation changes have even been found between hippocampus and blood cells[88]. The role of behavioral activity-dependent DNA methylation changes for human brain aging and the specificity of the effect for particular brain regions will be exciting research topics for the future.

ENR has an enormous potential to prevent or counteract brain dysfunctions in aged animals, including synaptic plasticity, hippocampal neurogenesis, and cognitive abilities[21,22,89]. Using genome-wide DNA methylation sequencing at single-nucleotide resolution, we here demonstrated that ENR also restores a substantial number of age-related DNA methylation changes in the hippocampus. Our data show the stimulating effects of ENR at the epigenetic level and provide a potential mechanism of how active interaction with the environment supports and promotes brain function throughout the lifespan.

## Methods

**Animals and ENR**. Female C57BL/6JRj mice were ordered from Janvier Labs and maintained on a 12 h light/dark cycle at a temperature of 23 °C ± 1 °C with 40–60% humidity at the animal facility of the Center for Regenerative Therapies Dresden. Food and water were provided ad libitum. At an age of 6 weeks, mice were randomly distributed to an enriched environment or control cages. The enriched environment was a 0.74 m² enclosure equipped with tunnels and plastic toys. Toys were rearranged once per week, but not in the last 4 days before analysis. In every experiment, ten mice were housed together in an enriched environment at the same time. Control mice stayed in standard polycarbonate cages (Type II, Tecniplast) in groups of five mice per cage. Dirty bedding material and toys were replaced once per week. The entire enriched environment was cleaned once per month. All experiments were conducted in accordance with the applicable European and national regulations (Tierschutzgesetz) and were approved by the local authority (Landesdirektion Sachsen). Mice in the enriched environment did not receive special food.

**Reduced representation bisulfite sequencing (RRBS)**. Genomic DNA was isolated from micro-dissected dentate gyrus tissue using the QIAamp DNA Micro Kit or the Allprep DNA/RNA Micro Kit (both QIAGEN) following the manufacturer's manuals. RRBS libraries were prepared using the Premium RRBS Kit (Diagenode) and purified twice using Agencourt AMPure XP beads (Beckman Coulter; 1X bead volume). Quality and concentration of RRBS libraries were determined using the high sensitivity NGS fragment analysis kit (Advanced Analytical) and a fragment analyzer with the capillary size of 33 cm (Advanced Analytical). Sequencing was performed using a HiSeq2500 (experiment with middle-aged animals; Fig. 1) or a Nextseq (experiment with young and aged animals; Figs. 2–5) platform in a 75 bp single-end mode with a minimum sequencing depth of 10 million reads per sample.

**Targeted bisulfite sequencing**. Genomic DNA was isolated from micro-dissected dentate gyrus tissue using the QIAamp DNA Micro Kit (QIAGEN) and bisulfite conversion was performed with 500 ng of DNA using the EZ DNA Methylation-Lightning Kit™ (Zymo Research). Genomic regions were amplified (31 cycles) from 50 ng of bisulfite converted DNA using Platinum™ Taq DNA Polymerase High Fidelity (Thermo Fisher Scientific) according to the manufacturer's recommended protocol. Primers for targeted bisulfite sequencing were designed using the online tool MethPrimer 1.0 with gene-specific sequences downloaded from the Ensembl genome browser. Primer sequences are depicted in Supplementary Data 9. DNA methylation profiles of Npas4 (Fig. 1h), were obtained by Sanger sequencing of at least ten individual clones per mouse. Briefly, PCR products were purified from agarose gels (QIAquick PCR Purification Kit, QIAGEN), ligated into pCR2.1 (Thermo Fisher Scientific) and plasmids were transformed into chemically competent E. coli TOP10. Plasmids were isolated using GeneJET Plasmid Miniprep Kit (Thermo Fisher Scientific) and inserts sequenced by Sanger sequencing from the M13 promoter of pCR2.1. For targeted bisulfite sequencing of Tiam1 and Tshz2 (Fig. 5), PCR products for one replicate were pooled at equal molarity, and libraries prepared using the Nextera DNA Library Preparation Kit (Illumina). Amplicons were sequenced using the MiSeq System (Illumina). Data were analyzed as described for RRBS.

**Quantitative RT-PCR**. RNA was isolated from micro-dissected dentate gyri using RNeasy Micro Kit (QIAGEN) and 500 ng were used for cDNA synthesis with Superscript™ II Reverse Transcriptase (Thermo Fisher Scientific). Quantitative PCRs were performed using QuantiFAST SYBR Green PCR Kit (QIAGEN) and the CFX Connect™ Real-Time PCR Detection System (Bio-Rad). Primer sequences were designed using the online tool Primer 3 version 0.4.0 and are listed in Supplementary Data 9. Results were analyzed using the delta Ct method. Housekeeping genes Actb (Fig. 1) or Oaz1 (aging experiments, Fig. 5) were used for the normalization of sample concentrations.

**Chromatin immunoprecipitation**. Chromatin immunoprecipitation was performed using the iDeal ChIP-seq kit for Transcription Factors (Diagenode) according to the manual. Micro-dissected dentate gyri from three animals were pooled for one chromatin immunoprecipitation. Briefly, tissue was homogenized using a Dounce homogenizer (five strokes) and fixed in 1% formaldehyde for 7 min at room temperature. Fixation was quenched with glycine and cells were lysed using the lysis buffers provided in the kit. Chromatin was fragmented to a size of 100–600 bp by sonication in 130 μl-microTUBE AFA Fiber tubes (Covaris) with a Covaris S2 Focused-ultrasonicator (6 min; 2% duty cycle; intensity 4; 200 cycles per burst). Successful shearing was confirmed by agarose gel electrophoresis. Each chromatin sample was incubated with 8 μg of Mecp2 antibody (Diagenode, pAb-052-050) for 12 h at 4 °C. The specificity of this antibody had been confirmed previously[61]. Inputs (1%) were removed before immunoprecipitation from each sample and incubated in the absence of the Mecp2 antibody. As a negative control, an additional sample was incubated with IgG antibody (provided in ChIP kit; Diagenode). Precipitated chromatin was de-crosslinked and purified with IPure beads v2 as described in the manual. DNA was eluted in 20 μl Buffer C from which 3 μl were used for every PCR. Quantitative PCR was performed with QuantiFAST SYBR Green PCR Kit (QIAGEN) and region-specific primers (Supplementary Data 9). Enrichment (% input) was calculated as $2^{((Ct_{input}-6.64-Ct_{sample}))} \times 100\%$.

**Bioinformatic data analysis**

*Calculation of differential DNA methylation.* Fastq reads were trimmed using Trim Galore 0.4.4 and the function *Cutadapt* 1.8.1. To remove cytosines that were filled in during end preparation, an additional 2 bp were cut off from every sequence with a detected adapter. Trimmed reads were mapped against mm10 using Bismark 0.19.0[90].

Global DNA methylation levels are the means of the methylation percentages over all cytosines that were covered by RRBS with at least ten reads in all samples of the respective experiments. Separation into the cytosine contexts CpG and CpH (CHH and CHG) was performed based on context annotations extracted from Bismark.

Detection of differentially methylated cytosines was performed using methylKit 1.5.2[91]. Briefly, methylation levels were extracted from sorted Binary Alignment Map files using the function *processBismarkAln*. Data were filtered for cytosines with a minimum coverage of ten reads and maximum coverage of 99.9% percentile using the function *filterByCoverage*. Using the function *unite*, cytosines were selected that were sufficiently covered in at least three samples per group. Differentially methylated cytosines were identified with the *methDiff* function with default settings applying the chi-squared test, multiple testing correction using the SLIM method, a significance threshold of $q < 0.001$, and a threshold for absolute DNA methylation differences larger than 25%. Detection of differentially methylated cytosines was performed separately for CpGs and CpHs.

Differentially methylated regions were calculated using the R package DSS v.2.38.0 using Wald test for beta-binomial distributions with the function callDMR, an adjusted *p*-value threshold of 0.05, a minimum of 3 CpGs, and an absolute methylation difference threshold of 25%.

Differentially methylated cytosines were annotated to the gene with the nearest transcription start site using data tables downloaded from Ensembl BioMart (as of 01.05.2019.)[92]. Genes with at least four annotated differentially methylated cytosines were considered differentially methylated genes. Gene names used in this study are Ensembl gene names. Corresponding Entrez identifiers were retrieved from the Bioconductor package AnnotationDbi[93] using Ensembl gene identifiers as keys. Mapping of mouse genes to homologous human genes was performed using data tables downloaded from Ensembl BioMart (as of 01.052019.)[92]. Images depicting the relative position of DNA methylation loci within a gene were downloaded from the UCSC genome browser (NCBI Refseq track) and are not scaled.

*Genomic feature annotation.* Genomic coordinates of CpG islands, exons, and introns were downloaded from the mouse genome (mm10) track of the UCSC genome browser[94]. CpG island shores were defined as 2 kb upstream and downstream of a CpG island. Promoter regions were determined as ±1 kb from the transcription start sites of all known transcripts. Genomic coordinates for enhancers in the mouse genome (mm10) were downloaded from the Ensembl Regulatory Build track of the UCSC genome browser[95]. Superenhancer regions are from the adult mouse cortex and were retrieved from dbSUPER[96]. Overlaps of cytosines with genomic regions or other cytosines were performed using the functions *subsetByOverlaps* or *findOverlaps* of the R package Genomic Ranges 3.7[97]. Genomic feature enrichment was analyzed using count tables of feature coverage of differentially methylated cytosines and all cytosines covered by RRBS (background). Odds ratios and *p*-values of feature enrichment were calculated using a logistic regression model and the function *glm* in R. Odds ratios are the exponential coefficients of the model. *P*-values were adjusted using false discovery rate (FDR).

*Transcription factor binding analysis.* The number of differentially methylated cytosines that overlap with position weight matrices of transcription factor binding motifs was determined using the *Biostrings* package with a minimum match score of 90%. Transcription factor motifs were retrieved from motifDb[98]. Motif enrichment for each transcription factor was tested by applying hypergeometric tests using the function *phyper(q−1, m, n, k, lower.tail = FALSE)* with q = number of differentially methylated cytosines overlapping with the motif, m = number of background cytosines overlapping with the motif, n = number of background cytosines not overlapping with the motif and k = total number of differentially methylated cytosines. Multiple testing correction of *p*-values was performed using the FDR method. Mecp2 motifs were retrieved using the package BSgenome.

For confirmation of Mecp2 binding sites, Mecp2 ChIP-Seq data were downloaded from GEO (GSE67293) and overlapped with dmCpGs using Genomic Ranges 3.7. A hypergeometric test for the enrichment analysis was performed as described for the motif analysis.

*Functional gene enrichment analyses.* Enrichment for GO cellular components was analyzed using the online tool GREAT[99] with positions of differentially methylated CpGs as query list and all covered CpGs as the background list. Pathway analysis was performed using the R package ReactomePA[100] with a minimum of 5 and a maximum of 300 genes per pathway. Enrichment for genes involved in synaptic processes was performed using the SynGO online tool (https://www.syngoportal.org/) with default settings. Enrichment analyses were performed with differentially methylated genes as query lists and all genes covered by RRBS as background lists. GO terms, pathways, and SynGO terms were considered "enriched" at a significance threshold of adjusted $p < 0.05$.

Gene set enrichment was analyzed by performing hypergeometric tests using the function *phyper(q−1, m, n, k, lower.tail = FALSE)* in R with q = number of differentially methylated genes overlapping with the gene set, m = number of background genes overlapping with the gene set, n = number of background genes not overlapping with the gene set and k = total number of differentially methylated genes. Genes involved in adult hippocampal neurogenesis were downloaded from MANGO v3.2[14]. The gene list for genes with a reported positive or negative effect on neurogenesis after protein manipulation or gene manipulation (in total 283 genes). Genes or cytosines with known aging-induced DNA methylation changes were extracted from[67,101,102] for peripheral tissues and from[36,53] for the hippocampus.

Overlap of sets of differentially methylated genes (Fig. 5a, Supplementary Fig. 10a) was analyzed using hypergeometric tests applying the function *phyper (q−1, m, n, k, lower.tail = FALSE)* in R with q = number of genes that overlap between both gene sets, m = number of genes in the gene set 1, n = number of background genes minus m and k = number of genes in the gene set 2. The significance of pathway overlap (Supplementary Fig. 3c) was calculated using a hypergeometric test with Reactome pathways presented in Supplementary Data 2 and 4 and considering a total number of 1653 annotated mouse Reactome pathways (as of 01.10.2020).

The enrichment map was generated according to Merico et al.[57] using genes that contain at least four differentially methylated cytosines (dmCpGs or dmCpHs). GO biological process, GO cellular component, Reactome, and Wikipathways enrichment analyses were performed in g:Profiler (https://biit.cs.ut.ee/gprofiler) using gene ensembl identifiers. The results from SynGO and MANGO enrichment analysis were integrated into the .gmt file downloaded from g:Profiler.

The network was generated using the app Enrichment Map 3.2.0 in Cytoscape 3.7.1 using a *q*-value threshold (FDR) of 0.05. Clusters were manually annotated. Groups with fewer than six connected terms were not drawn. The STRING protein interaction network was generated inside cytoscape 3.7.1 using human ensembl identifiers with the plugin STRING (protein query) at a confidence score cutoff of 0.5 with automated enrichment analysis.

**Analysis of adult hippocampal neurogenesis.** Mice were anesthetized with 100 mg/kg ketamine (WDT) and 10 mg/kg xylazin (Serumwerk Bernburg AG) and transcardially perfused with 0.9% sodium chloride. Brains were removed from the skull and one hemisphere fixed in 4% paraformaldehyde prepared in phosphate buffer (pH 7.4) overnight at 4 °C. Brains were incubated in 30% sucrose in phosphate buffer for 2 days and cut into 40 μm coronal sections using a dry-ice-cooled copper block on a sliding microtome (Leica, SM2000R). Sections were stored at 4 °C in cryoprotectant solution (25% ethyleneglycol, 25% glycerol in 0.1 M phosphate buffer, pH 7.4).

For immunofluorescent stainings, sections were washed and unspecific binding sites were blocked in phosphate-buffered saline supplemented with 10% donkey serum (Jackson Immuno Research Labs) and 0.2% Triton X-100 (Carl Roth) for 1 h at room temperature. Primary antibodies were applied overnight at 4 °C as follows: polyclonal goat anti-doublecortin (1:250; Santa Cruz, sc-8067), polyclonal rat anit-Ki67 (1:500; eBioscience, 14-5698-82), polyclonal goat anti-Sox2 (1:500; Santa Cruz, sc-17320). The secondary antibodies Cy3 Donkey Anti-Goat IgG and Alexa Fluor 488 Donkey Anti-Rat IgG (both 1:1000; Jackson ImmunoResearch) were incubated for 2 h at room temperature. Antibodies were diluted in phosphate-buffered saline supplemented with 3% donkey serum and 0.2% Triton X-100. Nuclei were labeled with 4′,6-diamidino-2-phenylindole (DAPI; 1:4000; Jackson Immuno Research Labs). Sections were mounted onto glass slides and cover-slipped using Aquamount (Polysciences Inc.). Stainings were imaged using a Zeiss Apotome equipped with an AxioCam MRm camera and the software AxioVision 4.8 (Zeiss). Cells were quantified on every sixth section along the rostro-caudal axis of the dentate gyrus.

**Statistics.** The statistical tests used are reported in the specific results or methods section. All *t*-tests were two-sided and performed using GraphPad Prism 6.0. When performing multiple comparisons, *p*-values were adjusted using FDR corrections. All measurements were taken from distinct samples. All presented summary graphs show means ± standard errors of the mean.

**Reporting summary.** A reporting summary for this article is available as a Supplementary Information file.

## Data availability
Sequencing data were deposited at GEO ("GSE138368"). Figures 1, 2, 4, and 5 contain associated raw data. Other raw data supporting the findings of this study can be found in the Source Data, Supplementary Data, or in the Figures throughout the manuscript, or from the corresponding author upon reasonable request. Source data and Supplementary Data are provided with this manuscript.

## Code availability
No new code was developed for this study. Existing data analysis packages for bioinformatic analyses were used in the programming language R as described in detail in the manuscript. Scripts for the use of these packages are available online (https://www.bioconductor.org/about/) or from the authors upon request.

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

## Acknowledgements
This study was financed from basic institutional funds from Technische Universität Dresden and Helmholtz Association and by the AMPro Consortium of the Helmholtz Association. S.Z. was supported by an EMBO Short-Term Fellowship and a fellowship from the International Max Planck Research School on the Life Course, Berlin. The authors thank Hongjun Song for the discussion and methodological support and Tomohisa Toda, Robert W. Williams, and Khyobeni Mozhui for their helpful comments on the manuscript.

## Author contributions
Conceptualization: S.Z. and G.K.; investigation/visualization: S.Z.; methodology/formal analysis: S.Z. and R.W.O; software/data curation: R.W.O., M.L., and S.Z.; resources: A.D. and G.K.; writing original draft: S.Z.; writing review and editing: S.Z., R.W.O., and G.K.

## Funding

## Competing interests
The authors declare no competing interests.
