## [Peer Review File · Nature Communications]

Reviewers' comments:

Reviewer #1 (Remarks to the Author):

This study presents the seductive thesis that environmental enrichment counters the effects of aging. The temptation is to accept the results because we want them to be true, and the work is very nicely presented as well. There are several aspects of this that require a more critical look though, and on reflection the data does not support the conclusions.

The first is the question of what environmental enrichment is, and whether it is relevant to human. For a start his should say in the title that it is a mouse study, and furthermore it should address at least briefly the extensive and complex literature on environmental enrichment in mouse behaviour, as well as the rationale for studying it via DNA methylation in dentate gyrus. The dentate gyrus does make sense given the proposed role of adult neurogenesis, but the comparative analysis with human uses human prefrontal cortex samples. More importantly though we do not know, nor do the authors critically examine whether the particular enriched environment they use has a cognitive or behavioural effect and whether whatever effects it does have come through differences in activity, feeding, or anything else.

The second crucial question is whether the analysis demonstrates anything to do with rejuvenation, and of this I have serious doubts. The main analysis is not reported in enough detail to be sure of what has actually been done*, but the main analysis of RRBS sequencing of DNA from dentate gyrus in enriched versus standard housing yields over 11000 differences, reportedly 1.25 % of covered CpGs. From figure 1 we can see that the criteria are 25% difference and $q < .0001$, so we can guess that the statistical test was some kind of t-test and probably FDR, and we can't be sure of the number of mice in each group or whether each mouse was RRBS analysed separately (I did not go sleuthing in the sequence files which might reveal some of this). These thresholds are not justified. This is a very large number of putative differences even at quite a small Q-value so one would want to know why this is. Do the authors think the biological effects of enrichment are so extreme as to cause this number of differences? Or are there perhaps unconsidered systematic differences inflating the statistics? In any event what follows is the kind of enrichment analysis of long lists now traditional in genomics studies without a clear result.

The authors then derive an even longer list of DNA methylation differences between standard-housed mice at two different ages. The centrepiece analysis then compares these differences with the enrichment vs standard housing ones to find a large fraction of them go the opposite way. This then leads to the claim that enrichment 'counters' aging or even (according to the title) 'rejuvenates' the brain. Even if the analysis is solid and correct, I think that the rejuvenation claim is an over-interpretation, but I do not accept the analysis at all. There is a null hypothesis that the authors do not adequately reject or even address at all. This is that if most of the DNA methylation differences in one or both of their comparisons were simply noise, then by chance they would be opposite half the time. Given the great length of the lists of differences that they are comparing this is by far the most likely explanation.

One counter argument to this might be that the enrichments shown in fig 1 d-g seem to show that the results of the standard vs enrichment comparison at least demonstrate specificity. I am not convinced by this, partly because the experimental design and statistical analysis are not presented in detail. More importantly experiments are of considerable technical difficulty

and the dentate gyrus must be a few milligrams at most of wet tissue. If the data were dominated by noise derived from the possibly varying selection of cell types recovered in each microdissection, then the apparent group differences would be enriched in functions relevant to that tissue.

In conclusion a much more rigorous analysis is required and even then it is sadly unlikely that this data set could reasonably be used to claim 'rejuvenation'.

Reviewer #2 (Remarks to the Author):

Zocher et al. expose female mice to environmental enrichment starting at 6 weeks of age for 3 months and study DNA-methylation in the dentate gyrus (DG). The authors observe substantial changes in DNA-methylation comparing ENR to corresponding control mice raised via standard housing conditions. They then performed a similar analysis in 6.5 week old mice subjected to 4 days of EE or 12 months of ENR (14 month old mice = aged). While DNA-methylation substantially differs amongst 6.5 and 14 month old mice, ENR appears to partially ameliorate this phenotype. DNA-methylation changes are observed across the entire genome and thus also overlap with Mesp2 binding sites. The changes also involve genes linked to adult neurogenesis. In line with previous data adult neurogenesis is increased in mice upon ENR. The study addresses a timely and important topic. Having this said, I think some more mechanistic insight will be needed before the study can be published in Nature Communications. Although the authors present a comprehensive analysis, the finding that aging and ENR alters DNA-methylation in the brain is not new. Similarly, the finding that ENR increases neurogenesis is also well-established. It would be important in my view to link these observations mechanistically.

In addition I have some other suggestions/questions to the authors

- I wonder if the substantial changes in DNA-methylation observed amongst groups might also be due to the fact that the "young" control group is only 6 or 6.5 weeks old. In my view these are not young but rather juvenile mice. Supplementary figure 4a seems to depict the overall distribution of DNA-methylation but no statistical analysis amongst groups.

- Do 14-month old mice show any brain-related impairments other than neurogenesis?...when compared to 6.5 or 4.5 month old mice?

- Fig. 1: How do the authors go from 11101 differentially methylated CpGs to 373 genes used for GO analysis?

- Fig. 1: N = 3 for the DNA-methylation analysis appears to be rather low.

- Please show for selected genes the DNA-methylation patterns affected by ENR.

- DNA-methylation at different regions of a gene are linked to distinct functions. The authors should analyze this for the detected genes and perform the pathway analysis accordingly. They should also take into account if genes are hyper or hypomethylated.

- The authors need to confirm for selected genes altered DNA-methylation upon ENR in the various conditions

- I could not find the corresponding data showing that DNA-methylation changes comparing 6.5 week to 14 month old mice shows a 92% overlap to the datasets generated in ref (18, 28). It would be nice to present a corresponding supplemental figure.

- Fig 2d. The comparison of pathways affected by DNA-methylation to in response to ENR and during aging is interesting. I wonder if the conclusion drawn is correct since there is no statistical analysis to suggest that there is indeed a meaningful overlap. The point would be that any dataset from a DG might show a similar overlap.

- What are the consequences of altered DNA-methylation in in the various groups. Do they translate into gene-expression changes and if so via which mechanisms.

- Are ENR-induced DNA-methylation changes essential for increased adult neurogenesis?

- Lines 433-434: Is it sensible to annotate every diff-methylated cytosine to the nearest gene? What happens if the detected cytosine is 'extremely far' from the closest gene? It sounds more adequate to first check the distribution cytosine-gene distances (computing a histogram) and based on such distribution introduce some threshold for doing the annotation.

- Lines 434-435: It is not clear why authors chose to label a gene as "differentially methylated" if it contained N=4 methylated cytosines. Would results be substantially altered if N=3 or 6, for example?

- Line 441: Are Cpg, exon and intron UCSC tracks referring to Human or Mouse? Please indicate.

- Lines 450-451: Unless some technical detail is missing, it would be better if authors referred to the "general linear model with binomial distribution" simply as a "logistic regression model", which is what they used.

- Lines 458,473: "phyper" and "dhyper" refer to R functions implementing the CDF (cumulative distribution function) and PDF (probability density function), respectively, for the hypergeometric distribution. Given that PDF is the derivative of the CDF, it is wrong to add both of them together as expressed in lines 458 and 473. Please correct or give more details.

Reviewer #3 (Remarks to the Author):

The paper by Zocher et al., tests whether environmental enrichment in mice which is known to delay cognitive decline in aging also alters epigenetic changes that occur in aging. This is an important question since changes in DNA methylation are emerging to be tightly associated with aging and might provide a mechanism for linking ENR and slowing down of aging. The authors show robust changes in DNA methylation induced by ENR in the dentate gyrus in young animals, these changes in CpG sites seem to be mostly involving hypomethylation, they are enriched in enhancers and seem to be targeting MeCP2 binding sites. Interestingly, robust changes in DNA methylation occur in Aged mice which are involve

primarily hypomethylation. 31.73% of age-related changes in aged mice are counteracted by ENR. These changes overlap with changes induced by ENR at young and middle age, target enhancers and highly enriched in MeCP2 sites with enrichment for genes repressed by MeCP2. Interestingly, an enriched network is neurogenesis which prompted the authors to examine neurogenesis in the dentate gyrus showing a 60% increase in neurogenesis in the brain of animals exposed to life long ENR. Bioinformatics analysis of three data sets reveals a significant overlap with genes that are dysregulated in aged brains.

Critique

This is a fascinating study that convincingly shows that ENR has a specific impact on DNA methylation in the brain and that this might be involved in the protective antiaging effect of ENR in the brain. What is remarkable is the impressive enrichment in regulatory regions, the enrichment in MeCP2 targets as well as the overlap with age related genes in human brains. The study has also intriguing implications for humans and the potential value of such interventions in delaying age related cognitive decline in humans.

However, some of the statements in the paper are not sufficiently supported by confirmatory validating data.

a. The authors claim that the hypomethylation of MeCP2 targets leads to reduced binding of MeCP2? This however was not tested by direct demonstration of reduced binding of MeCP2 to these targets using ChIP seq with MeCP2 antibody in ENR and aged standard aged and young brains. This would have provided the evidence needed for claiming functional meaning for the DNA methylation changes.

b. The authors claim that the changes in MeCP2 targets leads to relief of silencing of gene expression of the MeCP2 targets which are normally silenced by MeCP2. This could be supported by either RNAseq of these animals or targeted QRT PCR of the targeted genes.

c. The authors claim in the discussion that ENR increased adult hippocampal neurogenesis through the lifespan but provide evidence only for aged animals. To support this claim, the authors need to show neurogenesis data at several timepoints in life.

d. In overlaps with human data direction of changes in gene expression are not indicated. Once these are determined in the mice it will be important to determine whether the changes occur in same direction in humans and aging mice in ENR “corrected” genes.

e. The authors say that 31.73% of all age related mCpG are counteracted by ENR. Who are the 70% CGs that don't get counteracted by ENR? What are their characteristics and what is the functional implication?

f. There is no validation by a different method of the changes in DNA methylation by pyrosequencing of target genes for example.

Reviewer #4 (Remarks to the Author):

The title and abstract provide an effective summary of several interesting and novel findings : i) environmental enrichment prevented the aging-induced CpG hypomethylation at target sites of the methyl-CpG-binding protein Mecp2 and i) bioinformatics functional enrichment study of differentially methylated CpGs neuronal plasticity, adult hippocampal neurogenesis and conversely age-related cognitive decline in the human brain. Overall, these findings will be interesting to neuroscientists, researchers who study epigenetic changes, and people who advocate lifestyle interventions for countering age related cognitive decline.

Limitation: small sample size underlying the results of Figure 1: methylation differences were assessed in only 3 mice per group. However, the mice were kept in identical conditions.

Fortunately, more mice were used for the interesting analysis in Figure 2 which demonstrate

that environmental enrichment protects the dentate gyrus from age-related CpG and CpH methylation changes.

Interesting functional validation data: fluorescent staining to detect new-born neurons in STD and ENR mice. Good write up (including introduction and discussion) and effective figures.

The statistical analysis appears to be sound. Expert bioinformatics analysis using appropriate software tools. Interesting enrichment analysis SynGO, MANGO. Interesting transcription factor motif analysis which implicates Mecp2 target sites.

Minor comments.

1) The abstract should contain a short sentence that explains the technical term "environmental enrichment".

2) Page 4, line 55. Mention sample sizes explicitly so that readers are aware of the main limitation of the results in Figure 1.

3) Figure 5 could perhaps be relegated to the Supplement if there are space limitations. I don't think a hairball of network interactions conveys a lot of information.

No further comments

Zocher et al. **“Epigenetic rejuvenation of the mouse hippocampus by environmental enrichment”**

We thank all reviewers for their interest in our results and for their constructive comments. We have carefully revised our manuscript and are convinced that addressing these comments has helped making our study stronger. In particular, we have now extensively validated our identified genomic regions with ENR-induced DNA methylation changes by performing further analyses, additional genome-wide sequencing as well as using targeted validation of candidate loci. Moreover, by applying bisulfite sequencing in combination with Mecp2-ChIP and transcriptional profiling, we now show that the age-related loss of methylation at Mecp2 motifs is associated with reduced Mecp2 binding and transcriptional alterations at selected loci. ENR housing of aged mice counteracted the DNA methylation change and rescued Mecp2 binding as well as transcriptional alterations in aged brains. These new results add further mechanistic insight into how environmental enrichment counteracts the age-related decline of brain function, and will certainly be the starting point for future investigations into the role of Mecp2 in brain aging.

Response to Reviewer comments (reprinted in blue font).

Reviewer #1

This study presents the seductive thesis that environmental enrichment counters the effects of aging. The temptation is to accept the results because we want them to be true, and the work is very nicely presented as well. There are several aspects of this that require a more critical look though, and on reflection the data does not support the conclusions.

The first is the question of what environmental enrichment is, and whether it is relevant to human. For a start his should say in the title that it is a mouse study, and furthermore it should address at least briefly the extensive and complex literature on environmental enrichment in mouse behaviour, as well as the rationale for studying it via DNA methylation in dentate gyrus.

We thank the reviewer for these useful comments. We now mention in the title that this is a mouse study and have changed it to *“Epigenetic rejuvenation of the mouse*

hippocampus by environmental enrichment". We re-wrote the introduction and added details about the enriched environment paradigm, its influences on mouse behaviour and why we focussed our analysis on DNA methylation. This new information can be found on page 3-4, lines 30-60.

We also discuss the relevance of environmental enrichment for humans on page 27, lines 543-558, which reads as follows:

"The paradigm of environmental enrichment is a highly reductionistic construct, which addresses relative phenotypical changes that can be ascribed to environmental, non-genetic effects. By using inbred mice the genetic factor is tightly controlled for. Exposure to ENR elicits a large number of effects on brain and behaviour and the idea of "environmental enrichment" has been translated to the human situation to describe the impact of environmental influences on brain development, cognition, and especially neurorehabilitation. Humans show substantial inter-individual differences in their behavioral activity and the ones with a physically and mentally active lifestyle and social engagements are more likely to maintain brain function and health in old age. They also show considerable genetic variability, however, so that the experimental ENR construct in animal studies is not identical to what can be observed and more loosely described as "enriched environment" in humans. While this is a critically important caveat, particularly relevant for judging effect sizes, the animal studies offer a perspective that is almost inaccessible in humans. Effectively, what the paradigm allows estimating is the impact of environment and activity, if there were no genetic variation. We are thus not making claims about precise equivalents of our findings under human conditions and not even under feral conditions of the animals but about patterns of change that might be conserved (albeit obscured) and principles of underlying mechanisms."

The dentate gyrus does make sense given the proposed role of adult neurogenesis, but the comparative analysis with human uses human prefrontal cortex samples.

The comparison with human data was performed using prefrontal cortex tissue, because – to our knowledge – suitable data from human hippocampus do not exist yet (or are not publicly available). We also want to emphasize that we did not claim that the observed effects are specific to the hippocampus. To discuss this point, we have added the following paragraph in the discussion on page 27, lines 558-569:

“In the present study, we found that ENR counteracted aging effects in mice at genes that are dysregulated with age-related cognitive decline in humans. Since, to our knowledge, no comparable molecular data exists yet for the human hippocampus, we compared genes dysregulated in human prefrontal cortex with our epigenetic data from mouse dentate gyrus. Whether ENR effects on DNA methylation patterns in aging are specific to the dentate gyrus is currently unknown. However, since ENR modifies neuronal activity in many brain areas, including the prefrontal cortex (Li et al., 2020), the epigenetic effects are likely comparable between different brain areas. Furthermore, there is evidence that age-related DNA methylation changes are not tissue-specific, such that similar DNA methylation changes have even been found between hippocampus and blood cells (Harris et al., 2020). The role of behavioral activity-dependent DNA methylation changes for human brain aging and the specificity of the effect for particular brain regions will be exciting research topics for the future.”

More importantly though we do not know, nor do the authors critically examine whether the particular enriched environment they use has a cognitive or behavioural effect and whether whatever effects it does have come through differences in activity, feeding, or anything else.

The enriched environment used in this study was essentially identical to the enriched environments used in most of our previous studies. For this exact setup, Garthe et al, 2016 has demonstrated a stimulating effect on hippocampus-dependent spatial navigation in the Morris watermaze and we had validated in other experiments its pro-neurogenic effect in the hippocampus (see for example Fabel et al, 2009). Moreover, our recent study (Körholz et al, 2018) showed that housing in a very similar enriched environment improved motor coordination, reduced activity in the open field and increased object exploration in the novel object recognition task.

To clarify that we have used a well-characterized set-up for environmental enrichment, we have added the following sentence to the manuscript on page 5, lines 71-73:

“For the particular ENR paradigm used here, our previous studies had confirmed the stimulating effect on brain plasticity and hippocampus-dependent memory and learning (Garthe et al., 2016; Körholz et al., 2018).”

Regarding the source of the effects of environmental enrichment: environmental enrichment provides a combination of stimuli that include sensory, cognitive, social and physical factors. Which specific aspect of environmental enrichment promotes brain plasticity and cognition is despite almost 60 years of research mostly unknown in the field, but has been reviewed extensively (including our recent review Kempermann, 2019). In the original manuscript, we had already discussed this point and proposed that ENR-induced neuronal activation through continuous novelty stimulation (rearrangement of toys and addition of new toys) might cause DNA methylation changes at neuronal activity-related genes. Such mechanisms would be in accordance with previously reported neuronal activity-induced DNA methylation changes in the dentate gyrus. The discussion for this point can be found in the manuscript on page 26, lines 522-540. Since mice in ENR did not receive special food compared to controls (which we now mentioned in the methods on page 29, line 589), we can exclude influences of the quality of the food.

The second crucial question is whether the analysis demonstrates anything to do with rejuvenation, and of this I have serious doubts. The main analysis is not reported in enough detail to be sure of what has actually been done*, but the main analysis of RRBS sequencing of DNA from dentate gyrus in enriched versus standard housing yields over 11000 differences, reportedly 1.25 % of covered CpGs. From figure 1 we can see that the criteria are 25% difference and $q < .0001$, so we can guess that the statistical test was some kind of t-test and probably FDR, and we can't be sure of the number of mice in each group or whether each mouse was RRBS analysed separately (I did not go sleuthing in the sequence files which might reveal some of this). These thresholds are not justified. This is a very large number of putative differences even at quite a small Q-value so one would want to know why this is. Do the authors think the biological effects of enrichment are so extreme as to cause this number of differences? Or are there perhaps unconsidered systematic differences inflating the statistics? In any event what follows is the kind of enrichment analysis of long lists now traditional in genomics studies without a clear result.

We have now carefully revised the Methods section and added detailed statistical information to all figure legends.

We have applied state-of-the-art bioinformatic techniques for the analysis of the RRBS data. For detection of differentially methylated cytosines, we had used the R package “methylKit”, which is widely used by the community (see for example Bou Sleiman et al., 2020; Ziller et al., 2018). The method applied in this package has been evaluated and compared to other techniques (Wreczycka et al., 2017) and is even recommended by sequencing providers such as Illumina (<https://emea.illumina.com/products/by-type/informatics-products/basespace-sequence-hub/apps/methylkit.html>), Diagenode and others. We followed the recommendations as described in the tutorials (<https://github.com/al2na/methylKit>; <https://bioconductor.riken.jp/packages/3.5/bioc/vignettes/methylKit/inst/doc/methylKit.html>) and carefully validated our analyses. The thresholds were not randomly chosen but are in accordance with recommendations by the ENCODE consortium and experts in the field (see for instance: Ziller et al., 2015). Additionally, we consider a threshold for absolute DNA methylation differences of 25 % as stringent, given that we were working with a complex tissue (and potentially inter-cellular differences in the response to ENR) and not with a synchronized cell line.

To further justify the selected threshold, we have plotted the distribution of the absolute methylation difference for all analyzed CpGs and CpHs (Figure R1), demonstrating that potential “background noise” lies below a threshold of 25 %. We have included plots showing such distributions for the aging and housing comparisons in Supplementary Figure 3a of the revised manuscript.

Figure R1: A threshold for absolute DNA methylation differences of 25 % (red arrows) removes background noise at CpGs and CpHs.

The extent of the DNA methylation changes induced by ENR (1.25 %) compares to what has been reported in previous studies. For instance, Guo et al, reported that 1.4 % of analyzed cytosines changed DNA methylation in the dentate gyrus after acute neuronal activation (Guo et al., 2011). Furthermore, Halder and co-workers

detected more than 55,000 differentially methylated regions in the mouse brain during a learning task (Halder et al., 2016). Additionally, transcriptomic studies showed gene expression changes at ca 15 % of all genes expressed in neurons in response to novel environment exploration (Jaeger et al., 2018). Considering these previous reports and the strong neurological effects of ENR, we think that the here detected 11,000 CpGs do not overestimate the effect of ENR on DNA methylation.

The authors then derive an even longer list of DNA methylation differences between standard-housed mice at two different ages.

It is established that aging is associated with pronounced, genome-wide DNA methylation changes in any tissue (Cole et al., 2017; Hahn et al., 2017; Masser et al., 2018). The extent of our detected age-related methylation changes in the dentate gyrus is very similar to what previous studies found in the hippocampus (Hadad et al., 2018; Masser et al., 2017). Indeed over 90 % of the age-related DNA methylation changes identified in our study have been reproduced by two independent studies (Hadad et al., 2018; Masser et al., 2017).

The centrepiece analysis then compares these differences with the enrichment vs standard housing ones to find a large fraction of them go the opposite way. This then leads to the claim that enrichment 'counters' aging or even (according to the title) 'rejuvenates' the brain. Even if the analysis is solid and correct, I think that the rejuvenation claim is an over-interpretation, but I do not accept the analysis at all. There is a null hypothesis that the authors do not adequately reject or even address at all. This is that if most of the DNA methylation differences in one or both of their comparisons were simply noise, then by chance they would be opposite half the time. Given the great length of the lists of differences that they are comparing this is by far the most likely explanation. One counter argument to this might be that the enrichments shown in fig 1 d-g seem to show that the results of the standard vs enrichment comparison at least demonstrate specificity. I am not convinced by this, partly because the experimental design and statistical analysis are not presented in detail.

We have now improved reporting of the statistical analyses and experimental design (please see Methods section and figure legends). Moreover, we have performed the

following additional analyses and experiments that demonstrate that our findings are not “simply noise”:

1. We included the proper statistical analysis that shows that there is a significant overlap between the age-and ENR-induced DNA methylation changes (Fig. 2b).
2. We performed a number of validation analyses which are presented in Supplementary Fig. 5 and described in the main text lines 190-221. For instance, by applying random sampling of CpGs, we show that the detected overlap between age- and ENR-induced changes is clearly above background. Second, we also detected significant overlaps when we used different subsets of aged STD mice for determining age- and ENR-induced DNA methylation changes, showing that the overlaps are not a result of noise in the aged STD group.
3. The reviewer argued that the “great length of the lists of differences” makes it likely that there is an overlap by chance. Our analysis suggests that we would also find a significant overlap between the age-and ENR-induced DNA methylation changes if we would increase the stringency threshold for a significant DNA methylation change to 50 %, which reduces the number of compared differences to 15 % but does not abolish the overlap (Figure R2).

Methylation difference of min 50 %: overlap of age- and ENR-dependent methylation changes

Figure R2: Significant overlap between age-and ENR-induced DNA methylation changes with a more stringent threshold for calling differentially methylated cytosines (absolute methylation difference 50 %). Hypergeometric tests for all comparisons: $p < 10^{-08}$. Figure refers to data presented in Fig. 2b.

4. We have successfully validated ENR- and aging-induced DNA methylation changes at selected candidate loci using targeted bisulfite sequencing of an independent cohort of mice. We present these results in Fig. 1h and Fig. 5.
5. We have reproduced the age-related DNA methylation changes at 65 % of gene loci by performing genome-wide sequencing of a separate cohort of young and aged mice (n = 8; results presented in Supplementary Fig. 10a). Moreover, we had already validated the aging-induced DNA methylation changes using published data sets (Supplementary Fig. 3).
6. We have reproduced the ENR effect on aging at gene loci by performing an additional genome-wide sequencing experiment (n = 8; Supplementary Fig. 10; Figure 5a).
7. We associated changes in DNA methylation with corresponding changes in Mecp2 binding and transcription at the same locus (Fig. 5).
8. ENR-induced differentially methylated CpGs are not found isolated throughout the genome but in clusters of differentially methylated regions (see for instance Supplementary Fig. 2), which would not be the case if they were “simply noise”.

Moreover, as mentioned by the reviewer, our findings show specificity:

9. We found specificity in the genomic distribution of the DNA methylation changes. ENR-induced differentially methylated cytosines were enriched at enhancers (Fig. 1c, Fig. 3b).
10. The genes affected by ENR-induced DNA methylation changes showed a remarkable specificity at neuronal plasticity- and synapse-related genes (Fig. 1d-g; Fig. 4a; Fig. 5b)
11. There is specificity in the direction of the methylation change in the various conditions. For instance, aging was characterized by predominant CpG hypomethylation but CpH hypermethylation (Fig. 2b).

More importantly experiments are of considerable technical difficulty and the dentate gyrus must be a few milligrams at most of wet tissue. If the data were dominated by noise derived from the possibly varying selection of cell types recovered in each microdissection, then the apparent group differences would be enriched in functions relevant to that tissue.

Because of the good visibility of its borders, the dentate gyrus can be reproducibly dissected from the hippocampus without any difficulty (see for example our JOVE video: <https://www.ncbi.nlm.nih.gov/pmc/articles/PMC4131911>). All micro-dissections for the present study were performed by the same researcher who had several years of experience with dentate gyrus micro-dissections. Furthermore, the dentate gyrus comprises up to approximately 70 % excitatory granule neurons. Given that we applied a DNA methylation threshold of 25 %, it is highly unlikely that DNA methylation changes come through differences in cell type composition from micro-dissections. Additionally, we now provide validation of DNA methylation changes at selected candidates from independent cohorts of micro-dissected mice.

In conclusion a much more rigorous analysis is required and even then it is sadly unlikely that this data set could reasonably be used to claim 'rejuvenation'.

We thank the reviewer for the constructive feedback and hope that the additional analyses, new experimental validation, improved reporting and modified text of our revised manuscript will convince the reviewer.

Reviewer #2

Zocher et al. expose female mice to environmental enrichment starting at 6 weeks of age for 3 months and study DNA-methylation in the dentate gyrus (DG). The authors observe substantial changes in DNA-methylation comparing ENR to corresponding control mice raised via standard housing conditions. They then performed a similar analysis in 6.5 week old mice subjected to 4 days of EE or 12 months of ENR (14 month old mice = aged). While DNA-methylation substantially differs amongst 6.5 and 14 month old mice, ENR appears to partially ameliorate this phenotype. DNA-methylation changes are observed across the entire genome and thus also overlap with Mecp2 binding sites. The changes also involve genes linked to adult neurogenesis. In line with previous data adult neurogenesis is increased in mice upon ENR. The study addresses a timely and important topic. Having this said, I think some more mechanistic insight will be needed before the study can be published in Nature Communications.

Although the authors present a comprehensive analysis, the finding that aging and

ENR alters DNA-methylation in the brain is not new. Similarly, the finding that ENR increases neurogenesis is also well-established. It would be important in my view to link these observations mechanistically.

We thank the reviewer for finding our study “timely and important”. Although we understand the desire for a concrete mechanistic link, we want to highlight that our study goes far beyond just showing that aging and ENR alter brain DNA methylation patterns. In recent years it became clear that age-related DNA methylation changes are fundamentally involved in genomic instability and tissue-wide functional decline with aging. Our study is, to our knowledge, the first one to show that environmental stimulation/ behavioural interventions can prevent and even reverse such age-related epigenetic alterations. We believe that this main finding is very exciting and will be followed mechanistically at different experimental scales (molecular, cellular, behavioural) in future studies.

For the revised manuscript, we have focused on the molecular mechanism downstream of DNA methylation and investigated the influence of aging and ENR on Mecp2 binding in the brain. Our new data identified an age-dependent loss of Mecp2 binding at differentially methylated loci overlapping Mecp2 motifs which was sensitive to ENR housing.

In addition I have some other suggestions/questions to the authors

1.1 I wonder if the substantial changes in DNA-methylation observed amongst groups might also be due to the fact that the “young” control group is only 6 or 6.5 weeks old. In my view these are not young but rather juvenile mice. Supplementary figure 4a seems to depict the overall distribution of DNA-methylation but no statistical analysis amongst groups.

We chose 6 week-old mice as the young control group, since this is the age when the dentate gyrus is firmly established and to keep the experiments consistent with previous environmental enrichment studies from our laboratory. From our integration of the sequencing results of the different experiments depicted in Supplementary Figure 5a, the age-related DNA methylation changes do not differ between 6-week-old and 4-month-old mice, suggesting that there are no major changes between the ages. We have now added the statistical analysis for that ($p = 0.54$, 2-way ANOVA

with post hoc Tukey test). To further address the reviewer's concern, we have performed an additional experiment and compared methylomes of 3-month-old mice with 17-month-old mice. Using this new experiment presented in Supplementary Figure 10a, we could reproduce 65 % of the age-related DNA methylation changes detected in our initial experiment.

1.2 Do 14-month old mice show any brain-related impairments other than neurogenesis?...when compared to 6,5 or 4,5 month old mice?

To address this point, we have added a sentence on page 8, lines 138-141, which reads as follows:

“Previous studies showed that 14-month-old mice exhibit age-related behavioral deficits, including reduced locomotor activity, increased anxiety-like behavior and decreased spatial memory (Singhal et al., 2020), as well as changes in brain synapse composition that relate to reduced neuron function (Cizeron et al., 2020).”

1.3 Fig. 1: How do the authors go from 11101 differentially methylated CpGs to 373 genes used for GO analysis?

We considered genes as being differentially methylated if they contained a minimum of four significantly differentially methylated cytosines. We have added this information to the respective figure legends (in addition to its description in the methods section). For a detailed explanation of the approach, please see answer to reviewer question 1.13.

1.4 Fig. 1: N = 3 for the DNA-methylation analysis appears to be rather low.

We agree with the reviewer. This initial experiment was supposed to be a pilot experiment to analyze whether our ENR paradigm leads to DNA methylation changes in the dentate gyrus before we continued with the analysis of the aged brains. We have now highlighted this fact in the main text on page 5, line 70. The validation experiment of the Npas4 candidate has been performed with N = 12 mice per group (Fig. 1h).

1.5 Please show for selected genes the DNA-methylation patterns affected by ENR.

We have included plots demonstrating gene-specific DNA methylation changes in Figures 1h and 5 as well as in Supplementary Figures 2.

1.6 DNA-methylation at different regions of a gene are linked to distinct functions. The authors should analyze this for the detected genes and perform the pathway analysis accordingly. They should also take into account if genes are hyper or hypomethylated.

We have repeated the Reactome pathway analyses and the MANGO enrichment analyses separately for hypomethylated and hypermethylated genes with DNA methylation changes in enhancers, promoters or gene bodies, respectively. This analysis was run for all ENR comparisons made in the paper. The results are provided in Supplementary Data 7.

The results show that specifically genes with DNA methylation changes in gene bodies were enriched in neuronal plasticity-related pathways. However, many of the genes with DNA methylation changes in enhancers also have known roles in neuronal plasticity. These conclusions can, however, be influenced by the number of genes in the analysis – the functional enrichment analyses can yield uninformative results if very few or very many query genes are considered. In general, no difference was found in pathway enrichment between hypomethylated and hypermethylated genes. We have described these results in the main text on page 16 lines 323-329. However, we feel that making this separation in the main figures would not gain insight but risks reducing the power of the analysis.

1.7 The authors need to confirm for selected genes altered DNA-methylation upon ENR in the various conditions

We have now validated DNA methylation changes at selected genes using targeted bisulfite sequencing of independent biological samples. The results are presented in Figures 1h and Figure 5.

1.8 I could not find the corresponding data showing that DNA-methylation changes comparing 6.5 week to 14 month old mice shows a 92% overlap to the datasets generated in ref (18, 28). It would be nice to present a corresponding supplemental figure.

We now included these results in form of Venn diagrams in Supplementary Fig. 3d.

1.9 Fig 2d. The comparison of pathways affected by DNA-methylation to in response to ENR and during aging is interesting. I wonder if the conclusion drawn is correct since there is no statistical analysis to suggest that there is indeed a meaningful overlap. The point would be that any dataset from a DG might show a similar overlap.

We do understand the reviewers concern, because we have considered this question before. However, while we do think that a default overlap of pathways for a given tissue might be a problem when analyzing RNA sequencing data, where one would only compare tissue-expressed genes, we think there should not be a problem for genome-scale DNA methylation data. The methylation data analysis does not select for DG-specific genes but rather compares all genomic regions (independent of expression or methylation status). The presented analysis compares data from two independent datasets and, therefore, should not be biased towards DG-specific pathways. A default pathway overlap would only happen, if the presented genes are the only genes in the dentate gyrus genome that are capable of methylation change in any situation, for which we believe there is no evidence.

We have now performed the statistical test and found a significant overlap of ENR- and age-dependent pathways (hypergeometric test: $p = 2.1E-12$; considering that 10 of the 17 ENR-dependent pathways overlap with the 47 age-sensitive pathways and considering the total number of mouse Reactome pathways of 1653). We included this information in the text on page 9, line 155.

1.10 What are the consequences of altered DNA-methylation in in the various groups. Do they translate into gene-expression changes and if so via which mechanisms.

To analyse the functional consequences of ENR-/aging-induced DNA methylation changes, have now performed gene expression analysis of selected candidates by qPCR and analyzed Mecp2 binding at the respective loci by Mecp2-ChIP-qPCR. For the candidates *Tiam1* and *Tshz2*, we show that age-related hypomethylation of Mecp2 target loci correlated with reduced Mecp2 binding and transcriptional activation. ENR counteracts the loss of DNA methylation, as well as the

corresponding change in Mecp2 binding and gene expression. These new results are presented in Fig. 5.

Additionally, we have integrated previously published RNA sequencing data from the dentate gyrus of ENR housed mice into our analysis (presented in Supplementary Fig. 11). We found that genes with both hyper- and hypomethylated gene bodies were enriched among genes up-regulated after ENR. This implicates a relationship between gene body methylation and transcription, but also suggests that the interplay is regulated independently at the level of individual loci. We have mentioned these results in the main text on page 22 lines 428-431.

1.11 Are ENR-induced DNA-methylation changes essential for increased adult neurogenesis?

We agree with the reviewer that this is an interesting question. To answer it, ideally site-specific CRISPR/Cas9-based rescue of DNA methylation patterns in aged mice to the levels observed in ENR mice would need to be performed. However, we would like to emphasize that this experiment is methodologically extremely challenging, and we feel it goes far beyond the scope of this current manuscript. The efficiency of CRISPR/Cas9-based manipulation of DNA methylation is very low and has hardly been investigated *in vivo*. Also, from our data, it is unclear, whether the effect could be re-created by manipulating isolated candidates. A less elegant approach to answer this question could be the manipulation of enzymes of the DNA methylation machinery. However, mice with global deletions of DNA methyltransferases or Tet enzymes show major neurodevelopmental and neuromuscular deficits (or in the case of DNMTs do not survive until maturity). Additionally, other studies suggested that manipulation of the whole DNA methylation pathway would already affect neurogenesis under baseline (non-stimulated) conditions, which would make it difficult to distinguish potential effects from environmentally stimulated increases in adult neurogenesis.

1.12 Lines 433-434: Is it sensible to annotate every diff-methylated cytosine to the nearest gene? What happens if the detected cytosine is 'extremely far' from the closest gene? It sounds more adequate to first check the distribution cytosine-gene

distances (computing a histogram) and based on such distribution introduce some threshold for doing the annotation.

Figure R3: Distance of ENR-induced differentially methylated CpGs (dmCpGs) to the nearest transcription start site (TSS). Figure refers to data presented in Fig. 1.

As suggested by the reviewer, we plotted the distance of differentially methylated cytosines to the nearest transcription start site (TSS; Figure R3). The vast majority of dmCpGs (72.45 %) were located within a 20 kb area to the transcription start site. Given the average size of mouse genes of 20-30 kb (Gabel et al., 2015), and the average distance between genes of 100 kb (own analysis), the vast majority of dmCpGs will be properly annotated. Annotation of rare distal inter-genic regions might be difficult. However, annotation of dmCpGs to the nearest TSS is state of the art in DNA methylation/ epigenetic research (see for example Ziller et al., 2014).

1.13 Lines 434-435: It is not clear why authors chose to label a gene as "differentially methylated" if it contained N=4 methylated cytosines. Would results be substantially altered if N=3 or 6, for example?

We had applied a dmCpG threshold for differentially methylated genes to improve potential functional implications of the results from the gene set enrichment analyses (we reasoned that genes affected by several dmCpGs would be more likely to be functionally affected). Additionally, functional gene enrichment analyses are less meaningful with very high gene numbers as inputs.

We have now plotted a histogram of dmCpGs per gene and performed the MANGO and Reactome pathway analyses additionally with N = 1, N = 3 and N = 6 dmCpGs per gene (Figure R4). While we find with each of these parameters a significant

enrichment at MANGO genes (Fig. R4c), no pathways are significantly enriched with a threshold of 6 dmCpGs (presumably due to the low number of genes). The pathway analysis with $N = 3$ dmCpGs looks very similar to the results of $N = \text{min } 4$ dmCpGs presented in Fig. 1.

Figure R4: Different cytosine thresholds for defining ENR-induced differentially methylated genes. **A**, While RRBS covered most genes with more than 6 CpGs (background), 65 % of genes with ENR-induced dmCpGs (blue bar) contained only one dmCpG. **B**, Significantly enriched pathways from Reactome pathway enrichment of genes with a minimum of 1 dmCpG (left) and a minimum of 3 dmCpGs (right). No significantly enriched pathways were found when enrichment analysis was performed with genes containing at least 6 dmCpGs (103 genes). **C**, Genes with known function in adult hippocampal neurogenesis as annotated in MANGO are enriched among ENR-induced differentially methylated cytosines independent of the applied dmCpG threshold. Depicted p -values are from hypergeometric tests. MANGO, Mammalian Adult Neurogenesis Gene Ontology. Figure refers to content presented in Fig. 1.

1.14 Line 441: Are Cpg, exon and intron UCSC tracks referring to Human or Mouse? Please indicate.

The UCSC tracks for locations of exons, introns, CpG island and enhancers were specific for the mouse genome build mm10. We have now specified this in the methods section.

1.15 Lines 450-451: Unless some technical detail is missing, it would be better if authors referred to the "general linear model with binomial distribution" simply as a "logistic regression model", which is what they used.

We have changed the text accordingly.

1.16 Lines 458,473: "phyper" and "dhyper" refer to R functions implementing the CDF (cumulative distribution function) and PDF (probability density function), respectively, for the hypergeometric distribution. Given that PDF is the derivative of the CDF, it is wrong to add both of them together as expressed in lines 458 and 473. Please correct or give more details.

In our study, we used hypergeometric distributions to test for over-representation of genes or transcription factor motifs. The function $phyper(q, m, n, k, lower.tail = FALSE)$ with (q = number of differentially methylated genes overlapping with the gene set, m = number of background genes overlapping with the gene set, n = number of background genes not overlapping with the gene set and k = total number of differentially methylated genes) gives us the p -value for the null hypothesis $P[X > x]$. However, to test $P[X \geq x]$, we either have to subtract 1 from the hit list by applying $phyper(q-1, m, n, k, lower.tail = FALSE)$ or add $P[X=x]$ by computing $dhyper(q, m, n, k)$. Hence, the equations $phyper(q, m, n, k, lower.tail = FALSE)+dhyper(q, m, n, k)$ and $phyper(q-1, m, n, k, lower.tail = FALSE)$ yield the same p -value. We had initially reported $phyper(q, m, n, k, lower.tail = FALSE)+dhyper(q, m, n, k)$ but appreciate that this might be less intuitive for readers. We therefore re-run all test with $phyper(q-1, m, n, k, lower.tail = FALSE)$ and changed the methods accordingly.

Reviewer #3

The paper by Zocher et al., tests whether environmental enrichment in mice which is known to delay cognitive decline in aging also alters epigenetic changes that occur in aging. This is an important question since changes in DNA methylation are emerging to be tightly associated with aging and might provide a mechanism for linking ENR and slowing down of aging. The authors show robust changes in DNA methylation induced by ENR in the dentate gyrus in young animals, these changes in CpG sites seem to be mostly involving hypomethylation, they are enriched in enhancers and seem to be targeting MeCP2 binding sites. Interestingly, robust changes in DNA methylation occur in Aged mice which are involve primarily hypomethylation. 31.73% of age-related changes in aged mice are counteracted by ENR. These changes overlap with changes induced by ENR at young and middle age, target enhancers and highly enriched in MeCp2 sites with enrichment for genes repressed by MeCP2. Interestingly, an enriched network is neurogenesis which prompted the authors to examine neurogenesis in the dentate gyrus showing a 60% increase in neurogenesis in the brain of animals exposed to life long ENR. Bioinformatics analysis of three data sets reveals a significant overlap with genes that are dysregulated in aged brains.

Critique

This is a fascinating study that convincingly shows that ENR has a specific impact on DNA methylation in the brain and that this might be involved in the protective antiaging effect of ENR in the brain. What is remarkable is the impressive enrichment in regulatory regions, the enrichment in MeCP2 targets as well as the overlap with age related genes in human brains. The study has also intriguing implications for humans and the potential value of such interventions in delaying age related cognitive decline in humans.

However, some of the statements in the paper are not sufficiently supported by confirmatory validating data.

We thank the reviewer for finding our study “fascinating” and appreciate the constructive comments.

a. The authors claim that the hypomethylation of MeCP2 targets leads to reduced binding of MeCP2? This however was not tested by direct demonstration of reduced binding of MeCP2 to these targets using ChIP seq with MeCP2 antibody in ENR and aged standard aged and young brains. This would have provided the evidence needed for claiming functional meaning for the DNA methylation changes.

We thank the reviewer for this suggestion. We have now performed Mecp2 ChIP of differentially methylated genes in young STD, aged STD and aged ENR housed mice. These results demonstrate an age-related reduction of Mecp2 binding at intragenic enhancers of *Tshz2*, *Cux2* and *Tiam1*, which is rescued by environmental enrichment of aged mice (Figure 5). These results are exciting, since they suggest a previously unknown role of Mecp2 in brain aging.

b. The authors claim that the changes in MeCP2 targets leads to relief of silencing of gene expression of the MeCP2 targets which are normally silenced by MeCP2. This could be supported by either RNAseq of these animals or targeted QRT PCR of the targeted genes.

We have now performed qRT PCR of the genes at which we found an age-related reduction of Mecp2 binding. We observed that, at some loci (*Tshz2* and *Tiam1*), reduced DNA methylation and Mecp2 binding correlated with increased transcription. In contrast, we did not detect gene expression changes with aging or ENR at *Cux2*, which was also differentially bound by Mecp2. These new results are presented in Figure 5. We discuss that the relationship between age-related changes in DNA methylation and gene expression is locus- and context-dependent.

c. The authors claim in the discussion that ENR increased adult hippocampal neurogenesis through the lifespan but provide evidence only for aged animals. To support this claim, the authors need to show neurogenesis data at several timepoints in life.

We have now included data showing increased adult hippocampal neurogenesis after three months and six months of continuous ENR housing (4-month-old and 7-month-old mice, respectively). The results of this analysis are now integrated into Fig. 4c.

d. In overlaps with human data direction of changes in gene expression are not indicated. Once these are determined in the mice it will be important to determine whether the changes occur in same direction in humans and aging mice in ENR “corrected” genes.

We have re-analyzed the overlap of our differentially methylated genes with the Alzheimer-related differentially methylated genes in humans and took the direction of the DNA methylation changes into account (only overlap of aging-induced hypomethylation/ENR-induced hypermethylation in our dataset with Alzheimers-related hypomethylation and vice versa). However, because the interaction between age-related DNA methylation changes and gene expression/ proteomics is not unidirectional and presumably context-dependent, we feel that, at this stage, taking direction of the gene expression/ proteomic changes into account does not improve this analysis. We adjusted the text to clarify that this overlap was supposed to ask whether the same genes are affected by age-related dysregulation in humans and mice. We have updated Figure 6, legend and main text accordingly.

e. The authors say that 31.73% of all age related mCpG are counteracted by ENR. Who are the 70% CGs that don't get counteracted by ENR? What are their characteristics and what is the functional implication?

We have now included an analysis of the genomic localizations and the functional enrichment of genes at which ENR did not significantly counteract aging effects. The results of this analysis are presented in Supplementary Fig. 8 and described in the main text on page 16, lines 314-322, and mentioned in the discussion.

f. There is no validation by a different method of the changes in DNA methylation by pyrosequencing of target genes for example.

We have now validated DNA methylation changes at selected genes using targeted bisulfite sequencing of independent biological samples. The results are presented in Figures 1h and Figure 5. We additionally validated gene-specific effects of ENR on aging using an additional genome-wide sequencing experiment (Fig. 5a).

Reviewer #4

The title and abstract provide an effective summary of several interesting and novel findings : i) environmental enrichment prevented the aging-induced CpG hypomethylation at target sites of the methyl-CpG-binding protein Mecp2 and i) bioinformatics functional enrichment study of differentially methylated CpGs neuronal plasticity, adult hippocampal neurogenesis and conversely age-related cognitive decline in the human brain. Overall, these findings will be interesting to neuroscientists, researchers who study epigenetic changes, and people who advocate lifestyle interventions for countering age related cognitive decline. Limitation: small sample size underlying the results of Figure 1: methylation differences were assessed in only 3 mice per group. However, the mice were kept in identical conditions. Fortunately, more mice were used for the interesting analysis in Figure 2 which demonstrate that environmental enrichment protects the dentate gyrus from age-related CpG and CpH methylation changes.

We thank the reviewer for the positive comments and finding our study “interesting and novel”. We agree that the sample size in Figure 1 is small. This initial experiment was supposed to be a pilot experiment to analyse whether our ENR paradigm leads to DNA methylation changes in the dentate gyrus before we went on to the aging analysis. We have now highlighted this fact in the main text. Moreover, the validation of the candidate *Npas4* was performed with a larger sample size of N = 11.

Interesting functional validation data: fluorescent staining to detect new-born neurons in STD and ENR mice. Good write up (including introduction and discussion) and effective figures.

The statistical analysis appears to be sound. Expert bioinformatics analysis using appropriate software tools. Interesting enrichment analysis SynGO, MANGO. Interesting transcription factor motif analysis which implicates Mecp2 target sites. Minor comments.

1) The abstract should contain a short sentence that explains the technical term "environmental enrichment".

We have changed the wording of the abstract to read:

“Using genome-wide DNA methylation sequencing, we here show that living in a stimulus-rich environment counteracted age-related DNA methylation changes in the hippocampal dentate gyrus of mice.” (page 2, line 13-15).

In addition, in the Introduction (where word limits are less constrained) we have added the descriptive text:

“In enriched environments, groups of rodents freely explore large cages equipped with frequently rearranged toys, providing physical, cognitive, sensory and social stimulation.”

2) Page 4, line 55. Mention sample sizes explicitly so that readers are aware of the main limitation of the results in Figure 1.

We have changed the text accordingly. It now reads as follows:

“To first investigate whether ENR changes DNA methylation patterns in the adult dentate gyrus, we performed a pilot experiment and kept female C57BL/6JRj mice in ENR or standard housing cages (STD) for three months starting at an age of six weeks. ... We performed genome-wide DNA methylation profiling on micro-dissected dentate gyrus tissue of three animals per group by reduced representation bisulfite sequencing (RRBS) (Boyle et al., 2012; Meissner et al., 2005).”

3) Figure 5 could perhaps be relegated to the Supplement if there are space limitations. I don't think a hairball of network interactions conveys a lot of information.

We feel this figure provides a helpful graphical summary and are not aware of space restrictions at this stage. We would defer a decision on this point to the editor in the case that space becomes a limiting issue.

No further comments.

Reviewers' comments:

Reviewer #1 (Remarks to the Author):

The authors have made extensive changes to the manuscript, including substantial new data and analyses. I remain very much against the claim that what they observe can be called 'rejuvenation', and I'm still not fully convinced by their main analysis comparing age and environment effect directions. The response to reviewer 3's point d doesn't seem adequate.

On balance though the work is of great interest and it is now probably well enough described for readers to make their own judgement.

Reviewer #3 (Remarks to the Author):

The authors have adequately addressed the comments of the reviewers by clarifying their results and discussion and by performing new analyses and validation studies. The new analyses strengthen the conclusions of this study. I have no further reservations and I believe that the manuscript will be a very important that would attract wide attention.

Reviewer #4 (Remarks to the Author):

Thank you for successfully addressing all of my suggestions.
No further comments.

Author response to reviewer comments

Reviewer #1 (Remarks to the Author):

The authors have made extensive changes to the manuscript, including substantial new data and analyses. I remain very much against the claim that what they observe can be called 'rejuvenation', and I'm still not fully convinced by their main analysis comparing age and environment effect directions. The response to reviewer 3's point d doesn't seem adequate.

On balance though the work is of great interest and it is now probably well enough described for readers to make their own judgement.

We thank the reviewer for finding our work interesting. In the revised manuscript, we have removed all mentions of the term "rejuvenation" from the title, abstract and main text, which we agree was an over-interpretation of the data presented.

We have double-checked our response to the previous comment d of reviewer 3, which seems appropriate in our eyes.

Reviewer #3 (Remarks to the Author):

The authors have adequately addressed the comments of the reviewers by clarifying their results and discussion and by performing new analyses and validation studies. The new analyses strengthen the conclusions of this study. I have no further reservations and I believe that the manuscript will be a very important that would attract wide attention.

We thank the reviewer for finding our study important.

Reviewer #4 (Remarks to the Author):

Thank you for successfully addressing all of my suggestions.
No further comments.

We thank the reviewer for the feed-back.